# Plain Transformers Can be Powerful Graph Learners

**Liheng Ma**[*]                                                                                   *liheng.ma@mail.mcgill.ca*
*McGill University & Mila - Quebec AI Institute*

**Soumyasundar Pal**                                                                          *soumyasundar.pal2@huawei.com*
*Huawei Noah's Ark Lab, Montreal*

**Yingxue Zhang**                                                                                *yingxue.zhang@huawei.com*
*Huawei Noah's Ark Lab, Montreal*

**Philip Torr**                                                                                       *philip.torr@eng.ox.ac.uk*
*University of Oxford*

**Mark Coates**                                                                                    *mark.coates@mcgill.ca*
*McGill University & Mila - Quebec AI Institute*

**Reviewed on OpenReview:** *https://openreview.net/forum?id=bEmDvPOfdv*

## Abstract

Transformers have attained outstanding performance across various modalities, owing to their simple but powerful scaled-dot-product (SDP) attention mechanisms. Researchers have attempted to migrate Transformers to graph learning, but most advanced Graph Transformers (GTs) have strayed far from plain Transformers, exhibiting major architectural differences either by integrating message-passing mechanisms or incorporating sophisticated attention mechanisms. These divergences hinder the easy adoption of training advances for Transformers developed in other domains. Contrary to previous GTs, this work demonstrates that the plain Transformer architecture can be a powerful graph learner. To achieve this, we propose to incorporate three simple, minimal, and easy-to-implement modifications to the plain Transformer architecture to construct our Powerful Plain Graph Transformers (PPGT): (1) simplified $L_2$ attention for measuring the magnitude closeness among tokens; (2) adaptive root-mean-square normalization to preserve token magnitude information; and (3) a simple MLP-based stem for graph positional encoding. Consistent with its theoretical expressivity, PPGT demonstrates noteworthy realized expressivity on the empirical graph expressivity benchmark, comparing favorably to more complicated alternatives such as subgraph GNNs and higher-order GNNs. Its empirical performance across various graph datasets also justifies the effectiveness of PPGT. This finding underscores the versatility of plain Transformer architectures and highlights their strong potential as a unified backbone for multimodal learning across language, vision, and graph domains.

## 1 Introduction

Transformers have achieved excellent performance across various domains, from language (Vaswani et al., 2017; Devlin et al., 2019; Brown et al., 2020) to vision (Dosovitskiy et al., 2021; Touvron et al., 2021a;b), and are well known for their reduced dependency on inductive bias as well as stronger flexibility and generalizability (Dosovitskiy et al., 2021). The recent success of transformer-based, multi-modal, large language models (LLMs) (OpenAI, 2024; Dubey et al., 2024) has also brought the once-unattainable dream

---

[*]Part of this work was conducted while LM was a visiting PhD student at the University of Oxford. The code is available at https://github.com/LiamMa/PPGT.

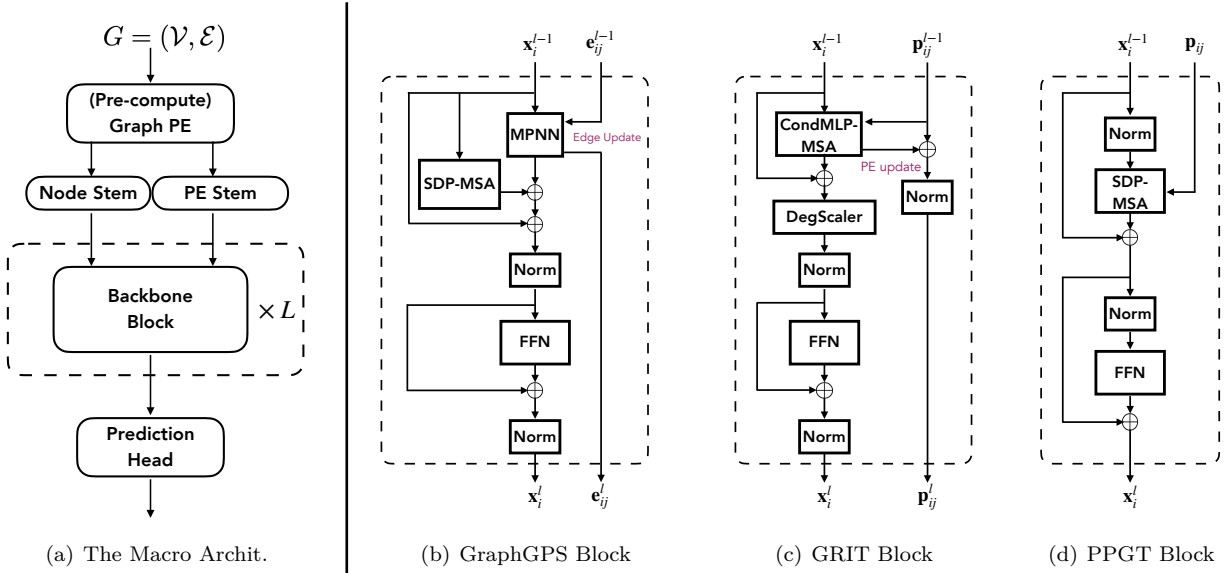

Figure 1: (a) Graph Transformers usually consist of preprocessing blocks (i.e., stems), backbone blocks (i.e., Transformer layers), and task-specific output heads. (b) GraphGPS introduces a complicated hybrid architecture integrating MPNN layers with sparse edge updates. (c) GRIT is equipped with a complex attention mechanism (conditional MLPs) with PE update and degree-scaler. On the other hand, (d) the proposed PPGT blocks simply follow the plain Transformer architecture, where $sL_2$ attention is implemented as SDP attention via float attention mask, and AdaRMSN is a direct substitute of RMSN.

of building artificial general intelligence closer to reality. However, these modalities are primarily data on Euclidean spaces, while non-Euclidean domains are still under-explored. Non-Euclidean domains like graphs are important for describing physical systems with different symmetry properties, from protein structure prediction (Jumper et al., 2021) to complex physics simulation (Sanchez-Gonzalez et al., 2018). Therefore, many researchers have attempted to migrate Transformer architectures for graph learning for the future integration into multimodal foundation models.

Nevertheless, unlike other Euclidean domains, the characteristics of graphs make the naive migration challenging. Following the suboptimal performance of an early attempt (Dwivedi & Bresson, 2021), researchers introduced significant architectural modifications to make Transformers perform well for graph learning. These include: implicit/explicit message-passing mechanisms (Kreuzer et al., 2021; Rampášek et al., 2022), edge-updating (Kim et al., 2022; Hussain et al., 2022; Ma et al., 2023), and sophisticated non-SDP (scaled-dot-product) attention designs (Chen et al., 2022; Ma et al., 2023). These complexities hinder the integration of advances for Transformers arising from other domains and impede progress towards potential future unification of multi-modalities.

In this work, we step back and rethink the difficulties that a plain Transformer architecture faces when processing graph data. Instead of adding significant architectural modifications, we propose three minimalist but effective modifications to empower plain Transformers with advanced capabilities for learning on graphs (as shown in Fig. 1). This results in our proposed **_Powerful Plain Graph Transformers_** (PPGTs), which have remarkable empirical and theoretical expressivity for distinguishing graph structures and superior empirical performance on real-world graph datasets. These modifications can be incorporated without significantly changing the architecture of the plain Transformer. Thus, our proposal both retains the simplicity and generality of the Transformer and offers the potential of facilitating cross-modality unification.

## 2 Preliminaries

### 2.1 Graph Learning

**Learning of Graphs and Encoding of Multisets** Graphs are non-Euclidean geometric spaces with irregular structures and symmetry to permutation (i.e., invariant/equivariant). A key factor for learning graphs is the ability to distinguish the distinct structures of the input graphs, which is typically referred to as the expressivity of the graph model (Xu et al., 2019; Zhang et al., 2023b). Currently, most graph neural networks (GNNs) are developed based on the framework of the Weisfeiler-Leman (WL) Isomorphism test (Weisfeiler & Leman, 1968) – a color-refinement algorithm based on multisets $\{\{\cdot\}\}$ encoding. For example, message-passing networks (MPNNs) based on 1-WL (Xu et al., 2019), distinguish graphs by encoding the neighborhood of each node as a multiset (Xu et al., 2019). To go beyond 1-WL, researchers have extended to $K$-WL GNNs (Morris et al., 2019), $K$-Folklore-WL (FWL) GNNs (Feng et al., 2023), and Generalized-distance-WL (GD-WL) GNNs (Zhang et al., 2023b), which are also based on such multiset encoding schemes, albeit with different multiset objects. It is worth mentioning that, as discussed in Xu et al. (2019); Zhang et al. (2023b), the cardinalities of multisets are crucial for distinguishing multisets, e.g., $\{\{a, b\}\}$ versus $\{\{a, a, b, b\}\}$. Generally, the cardinality is encoded into the token representation as its magnitude, e.g., encoding $\{\{a, b\}\}$ and $\{\{a, a, b, b\}\}$ as $\mathbf{x}$ and $c \cdot \mathbf{x}$ respectively, for $c \neq 1 \in \mathbb{R}^+$. Thus, the ***loss of token magnitude information weakens the ability to distinguish multisets and, consequently, graph structures***.

**Graph Transformers – Learning Graphs with Pseudo-coordinates** The philosophy of Graph Transformers (GTs) is to learn graph representations using positional encodings (PEs) rather than operating directly on the original input graphs. Zhang et al. (2023b) provide a theoretical framework (GD-WL) that demonstrates that GTs' stronger expressivity stems from finer-grained information beyond 1-WL encoded in graph PEs. Zhang et al. (2024b) demonstrate that GD-WL not only applies to distance-like or relative graph PEs but also enables analysis of absolute PEs (e.g., Laplacian PE) by transforming them into their relative counterparts. Concurrently, Ma et al. (2024) interpret graph PE as pseudo-coordinates in graph spaces that mimic manifolds, demonstrating that it can extend beyond graph distance and is not restricted to the Transformer architecture (e.g., convolutions on pseudo-coordinates). ***We follow the same philosophy and show that with a simple stem on pseudo-coordinates, even plain Transformers can achieve strong expressive power in graph learning.***

### 2.2 Limitations in Plain Transformer Architectures

**The Loss of Magnitude Information in Token-wise Normalization Layer** LayerNorm (LN (Ba et al., 2016)) and Root-Mean-Square-Norm (RMSN (Zhang & Sennrich, 2019)) are two widely used token-wise normalization techniques in Transformer-based models that effectively control token magnitudes:

$$\text{LN}(\mathbf{x}) = \frac{\mathbf{x} - \frac{1}{D}\mathbf{1}^\intercal\mathbf{x}}{\frac{1}{\sqrt{D}}\|\mathbf{x} - \frac{1}{D}\mathbf{1}^\intercal\mathbf{x}\|} \odot \boldsymbol{\gamma} + \boldsymbol{\beta}, \qquad \text{RMSN}(\mathbf{x}) = \frac{\mathbf{x}}{\frac{1}{\sqrt{D}}\|\mathbf{x}\|} \odot \boldsymbol{\gamma}. \tag{1}$$

Here $\mathbf{x} \in \mathbb{R}^D$ are token vectors; $\|\cdot\| \in [0, \infty)$ is the $L_2$-norm (i.e., magnitude) of a vector; and $\boldsymbol{\gamma}, \boldsymbol{\beta} \in \mathbb{R}^D$ are parameters of a learnable affine transform.

They retract token representations onto a hypersphere, a property essential for dot-product attention mechanisms, echoing the notion of retraction as used in computational physics. However, ***both LN and RMSN are strictly invariant to changes in input magnitude*** (see Proposition E.3 and the case study in Appx. C.2), which ***can result in the loss of valuable token magnitude information***

**Pitfalls of Scaled Dot-product Attention** With a good balance of capacity and efficiency, Scaled dot-product (SDP) attention has become the most common attention mechanism in modern Transformers (Vaswani et al., 2017). It has been widely explored in previous works and well optimized in deep learning libraries.

However, SDP attention is not perfect. For query and key tokens $\mathbf{q}_i, \mathbf{k}_j \in \mathbb{R}^D$, SDP attention is:

$$\alpha_{ij} := \texttt{Softmax}_j(\hat{\alpha}_{ij}) = \frac{\exp(\hat{\alpha}_{ij})}{\sum_{j'}\exp(\hat{\alpha}_{ij'})}, \text{where } \hat{\alpha}_{ij} := \frac{\mathbf{q}_i^\intercal\mathbf{k}_j}{\sqrt{D}} = \frac{\cos(\mathbf{q}_i, \mathbf{k}_j) \cdot \|\mathbf{q}_i\| \cdot \|\mathbf{k}_j\|}{\sqrt{D}}. \tag{2}$$

Here $\cos(\mathbf{q}_i, \mathbf{k}_j) \in [-1, 1]$ is the cosine similarity, measuring the angle between $\mathbf{q}_i$ and $\mathbf{k}_j$, independent of the vector magnitudes.

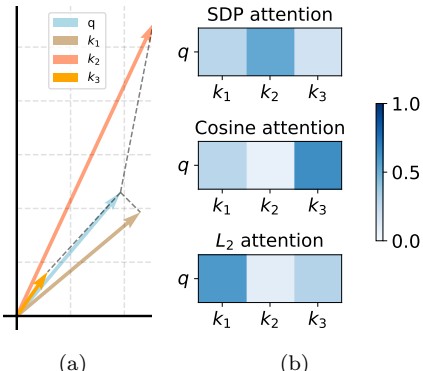

Figure 2: Illustration for comparing different attention mechanisms: (b) visualization of attention scores. SDP attention is biased towards larger-magnitude $k_2$. Cos attention disregards the magnitude information. $L_2$ attention strikes a balance between SDP and Cos attention to attend to $k_1$, which has the lowest $L_2$ distance to the query $q$.

Three drawbacks of SDP attention necessitate the additional control of token magnitudes:

1. ***Softmax saturation***: large token magnitudes (i.e., $\|\mathbf{q}_i\|$, $\|\mathbf{k}_j\|$) lead to large pre-softmax-logit values $\hat{\alpha}_{ij}$ and consequently extremely small gradients in backpropagation (Vaswani et al., 2017).
2. ***No closeness measurement on magnitude***: for each query, $\|\mathbf{q}_i\|$ degenerates to a temperature factor $\tau$, and only controls the sharpness of attention scores, irrespective of the closeness between $\|\mathbf{q}_i\|$ and $\|\mathbf{k}_j\|$ (Fig. 2).
3. ***Biased to large magnitude keys***: with $\|\mathbf{q}_i\|$ as the temperature $\tau$, compared to $\cos(\mathbf{q}_i, \mathbf{k}_j) \in [-1, 1]$, large $\|\mathbf{k}_j\| \in [0, +\infty)$ will dominate attention scores.

To mitigate the first and third drawbacks, ***existing plain Transformers heavily rely on token-wise normalization to regulate the token magnitudes***. Removing LN/RMSN and/or using BatchNorm(BN) (Ioffe & Szegedy, 2015) instead can induce training instability and divergence (Touvron et al., 2021b; Yao et al., 2021). While effective in language modeling, token-wise normalizations such as LN and RMSN are not without limitations. As shown in Appx. E.2 (theoretical analysis) and Appx. C.2 (empirical case study), LN and RMSN remove magnitude information, which can be crucial for downstream tasks.[1]

## 3 Method

As established in Section 2, plain Transformers exhibit suboptimal performance in graph learning, due to the architectural limitations that hinder preserving and modeling token magnitude information — a critical aspect of capturing graph structures. Consequently, previous GTs attempt to address it by adopting BN and overly sophisticated attention designs (e.g., MPNNs and/or MLP-based attention) (Kreuzer et al., 2021; Rampášek et al., 2022; Ma et al., 2023). They have thus strayed away from plain Transformers, impeding the transfer of previously explored training advances and the potential unification of other foundation models.

In this work, we propose two minimal and easy-to-adopt modifications to directly address the aforementioned limitations in plain Transformers. Furthermore, we introduce an extra enhancement in the PE stem to boost the information extraction of PE. These enhancements enable plain Transformers to achieve stronger expressive power for graph learning.

---

[1]This issue is less pronounced in language modeling but becomes significant in graph learning (Xu et al., 2019).

### 3.1 AdaRMSN

As discussed above, SDP attention mechanisms necessitate additional control on token magnitudes. However, existing token-wise normalization layers lead to irreversible information loss on magnitudes. We desire a normalization layer that can not only control the token magnitudes, but is also capable of preserving the magnitude information when necessary.

Inspired by adaptive normalization layers (Dumoulin et al., 2017; de Vries et al., 2017; Peebles & Xie, 2023), we propose adaptive RMSN (AdaRMSN)

$$\text{AdaRMSN}(\mathbf{x}) = \frac{\mathbf{x}}{\frac{1}{\sqrt{D}}\|\mathbf{x}\|} \cdot \gamma'(\mathbf{x}), \quad \text{where } \gamma'(\mathbf{x}) := \frac{1}{\sqrt{D}}\|\boldsymbol{\alpha} \cdot \mathbf{x} + \boldsymbol{\beta}\|. \tag{3}$$

The parameter $\boldsymbol{\beta} \in \mathbb{R}^D$ is initialized as $\mathbf{1}$ and $\boldsymbol{\alpha} \in \mathbb{R}^D$ as $\mathbf{0}$, leading to $\gamma'(\mathbf{x}) = 1$. AdaRMSN behaves the same as regular RMSN at the initial stage of training, but is capable of recovering the identity transformation with $\boldsymbol{\beta} = \mathbf{0}$ and $\boldsymbol{\alpha} = \mathbf{1}$ when necessary. [2] This initialization guarantees stable training in the early phases, comparable to regular RMSN, without requiring additional implementation details.

### 3.2 Simplified $L_2$ Attention Mechanisms

**From Dot-product to Euclidean Distance**  To empower SDP attention to sense both angle- and magnitude-information among query and key tokens, we revisit $L_2$ attention (Kim et al., 2021), which is based on Euclidean distance and is capable of measuring token closeness by balancing both angles and magnitudes, in contrast to SDP attention and cosine-similarity attention (as shown in Fig. 2).

To achieve better alignment with SDP attention, we further simplify it and reformulate it as SDP-attention with an additional bias term:

$$\alpha_{ij} := \texttt{Softmax}_j\left(-\frac{1}{\sqrt{D}} \cdot \frac{1}{2}\|\mathbf{q}_i - \mathbf{k}_j\|_2^2\right) = \texttt{Softmax}_j\left(\frac{1}{\sqrt{D}}(\mathbf{q}_i^\mathsf{T}\mathbf{k}_j - \frac{1}{2}\mathbf{q}_i^\mathsf{T}\mathbf{q}_i - \frac{1}{2}\mathbf{k}_j^\mathsf{T}\mathbf{k}_j)\right)$$

$$= \texttt{Softmax}_j\left(\frac{1}{\sqrt{D}}\mathbf{q}_i^\mathsf{T}\mathbf{k}_j - \frac{1}{2\sqrt{D}}\mathbf{k}_j^\mathsf{T}\mathbf{k}_j\right). \tag{4}$$

This can be easily supported by existing SDP attention implementations in most deep learning libraries, e.g., PyTorch (Paszke et al., 2019) [3].

We denote this attention mechanism as ***simplified $L_2$ (s$L_2$) attention***.

**s$L_2$ Attention with PE and Universality Enhancement**  For effective learning of structured objects, we need to inject positional encoding (PE) to enable the attention mechanism to sense the structure. Following previous work (Zhang et al., 2024b), we describe s$L_2$ attention using the relative-form of PE.

Let $\mathbf{p}_{ij}$ denote the relative positional embeddings for node-pair $(i, j)$ shared by all attention blocks, which is potentially processed by the stems (preprocessing modules). For each head, the attention scores $\alpha_{ij} \in \mathbb{R}$ for query/key tokens $\mathbf{q}_i, \mathbf{k}_j \in \mathbb{R}^D$ in the proposed attention are computed as

$$\alpha_{ij} := \phi(\mathbf{p}_{ij}) \cdot \texttt{Softmax}_j\left(\frac{\mathbf{q}_i^\mathsf{T}\mathbf{k}_j}{\sqrt{D}} - \frac{\mathbf{k}_j^\mathsf{T}\mathbf{k}_j}{2\sqrt{D}} + \theta(\mathbf{p}_{ij})\right), \tag{5}$$

where $\phi : \mathbb{R}^D \to \mathbb{R}$ and $\theta : \mathbb{R}^D \to \mathbb{R}$ are linear transforms. The $\phi(\mathbf{p_{ij}})$ is an optional term purely based on the relative position to guarantee the *universality of attention with relative PE* (URPE) (Luo et al., 2022). This term demands slight customization of the existing attention implementation, but we retain it since it is beneficial for learning objects with complicated structures (Luo et al., 2022; Zhang et al., 2023b). Notably, this form can be viewed as Continuous Kernel Graph Convolution (Ma et al., 2024) with a dynamic density function.

---

[2] We provide a stress test on AdaRMSN regarding the initialization in Appx. C.4
[3] The bias term can be directly formulated as a float attention mask in SDP-attention in PyTorch.

Note that, the attention with PE in relative-form is general, since it is widely employed in many existing Transformers, from language (Shaw et al., 2018; Raffel et al., 2020; Press et al., 2022) to vision (Dosovitskiy et al., 2021; Liu et al., 2021; 2022). Many absolute PEs are *de facto* explicitly/implicitly transformed into the relative-form in attention mechanisms (Su et al., 2024; Huang et al., 2024; Zhang et al., 2024b).

### 3.3 Powerful Plain Graph Transformers

In this section, we provide an example of constructing plain Transformers for learning graphs with our proposed techniques, termed Powerful Plain Graph Transformers (PPGT).

We follow the plain Vision Transformers architecture (Dosovitskiy et al., 2021; Touvron et al., 2021a), — stems, backbone and prediction head (as shown in Fig. 1(a) and Fig. 1(d)).

**Graph Positional Encoding**   In this work, we utilize relative random walk probabilities (RRWP) (Ma et al., 2023) as our running example of graph PE, considering its simplicity and effectiveness. RRWP is defined as

$$\mathbf{p}'_{ij} = [\mathbf{I}, \mathbf{M}, \mathbf{M}^2, \ldots, \mathbf{M}^{K-1}]_{[i,j]} \in \mathbb{R}^K \,, \tag{6}$$

where $\mathbf{X}_{[i,j]}$ stands for the $i,j$th element/slice of a tensor $\mathbf{X}$; $\mathbf{M} := \mathbf{D}^{-1}\mathbf{A}$ is the random walk matrix given the adjacency matrix $\mathbf{A}$ of the graph; and $\mathbf{I} \in \mathbb{R}^{N \times N}$ denotes the identity matrix. We follow the recipe in (Ma et al., 2023) – $p'_{ii}$ is incorporated into the node features after a linear projection; The pairwise terms $p'_{ij}$ (including $i = j$) are processed by a PE stem (described later) and injected into the attention mechanism as relative positional encodings. Most other graph PEs are applicable in our framework with minor modifications to the PE stem.

**Transformer Backbone**   Following the latest plain Transformers, we utilize the pre-norm (Xiong et al., 2020) architecture, for $l = 1, \cdots, L$,

$$\hat{\mathbf{X}}^l = \mathbf{X}^{l-1} + \texttt{MSA}(\texttt{Norm}(\mathbf{X}^{l-1}), \mathbf{P}), \quad \mathbf{X}^l = \texttt{FFN}(\mathbf{X}^l) := \hat{\mathbf{X}}^l + \texttt{MLP}(\texttt{Norm}(\hat{\mathbf{X}}^l)), \quad \mathbf{Y} = \texttt{Norm}(\mathbf{X}^L) \tag{7}$$

where $\mathbf{X}^l = [\mathbf{x}^l_i]_{i=1}^N \in \mathbb{R}^{N \times D}$ contains the node representations at layer $l$; $\mathbf{P} = [[\mathbf{p}_{ij}]_{i=1}^N]_{j=1}^N \in \mathbb{R}^{N \times N \times D}$ contains the relative positional embeddings; $\texttt{MSA}$ denotes multihead self-attention; $\texttt{Norm}$ indicates the normalization layer; $\texttt{MLP}$ is a 2-layer multilayer perception and $\texttt{FFN}$ denotes a Feedforward network with pre-norm.

**Stem**   Before the Transformer backbone, we use small networks, usually referred to as stems, to process positional encoding $\mathbf{p}'_{ij}$ and merge node/edge attributes $\mathbf{x}'_i/\mathbf{e}'_{ij}$.

We consider a simple stem design for node and PE, respectively:

$$\mathbf{x}^0_i = \texttt{FC}(\mathbf{x}'_i) + \texttt{FC}(\mathbf{p}'_{ii}), \quad \mathbf{p}^0_{ij} = \texttt{Norm} \circ \texttt{FFN} \circ \cdots \circ \texttt{FFN}(\texttt{FC}(\mathbf{e}'_{ij}) + \texttt{FC}(\mathbf{p}'_{ij})), \tag{8}$$

where $\texttt{FC}$ stands for the fully-connected layer (e.g., linear projection); $\circ$ stands for function composition; $\mathbf{x}'_i$ is treated as zero if there are no node attributes; $\mathbf{e}'_{ij}$ is set to zero if there are no edge attributes or if $(i, j)$ is not an observed edge. Driven by the analysis of RRWP by Ma et al. (2023), we introduce additional FFNs and a final normalization layer in the PE stem to better extract the structural information, mimicking the pre-norm architecture of the Transformer backbone.

**Prediction Head**   Unless otherwise specified, we employ a task-specific MLP prediction head, following the designs of GRIT (Ma et al., 2023) and GraphGPS (Rampášek et al., 2022) – for graph-level tasks, we apply sum or mean pooling followed by an MLP; for node-level tasks, we use an MLP shared across all nodes. For OGBN-ArXiv, we utilize a class-attention prediction head inspired by CaiT (Touvron et al., 2021b), which is better suited for the graph-sampling strategy.

### 3.4 Anti-Spectral-Bias with Sinusoidal PE Enhancement (SPE)

As discussed by Zhang et al. (2023b; 2024b), as the information provided through graph positional encodings (PEs) becomes more fine-grained, GD-GNNs can achieve better distinguishability of graph structure. However,

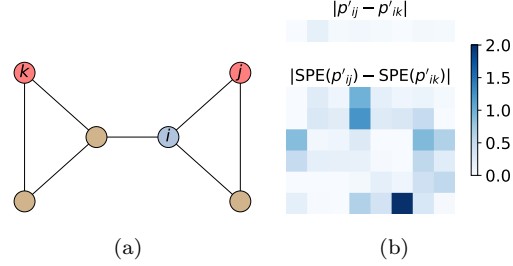

(a)                                   (b)

Figure 3: Illustration of two node-pairs (a) $(i, j)$ and $(i, k)$ of a graph, and (b) absolute difference of RRWPs and sinusoidally-encoded RRWPs for those two node-pairs.

due to the spectral bias of neural networks (Rahaman et al., 2019), MLPs prioritize learning the low-frequency modes and thus lose detailed information stored in PEs.

To mitigate this issue, motivated by NeRF (Mildenhall et al., 2020), we add an extra sinusoidal encoding on top of the RRWP and process it with a simple MLP. The sinusoidal encoding is applied to each channel of $\mathbf{p}'_{ij}$ in an elementwise fashion:

$$\mathtt{SPE}(p'_{ijk}) = \left[ p'_{ijk}, \sin(2^0 \pi p'_{ijk}), \ \cos(2^0 \pi p'_{ijk}), \ldots, \quad \sin\left(2^{S-1} \pi p'_{ijk}\right), \ \cos\left(2^{S-1} \pi p'_{ijk}\right) \right], \tag{9}$$

where $\mathtt{SPE} : \mathbb{R} \to \mathbb{R}^{1+2S}$ with $S \in \mathbb{Z}^+$ different bases. For notational conciseness, we use $\mathtt{SPE}(\mathbf{p}'_{ij}) \in \mathbb{R}^{K+2SK}$ to denote the application of $\mathtt{SPE}$ to all channels of $\mathbf{p}'_{ij}$ and concatenate the outputs. The PE stem (Eq. (8)) becomes

$$\mathbf{p}^0_{ij} = \mathtt{Norm} \circ \mathtt{FFN} \circ \cdots \circ \mathtt{FFN}(\mathtt{FC}(\mathbf{e}'_{ij}) + \mathtt{MLP}(\mathtt{SPE}(\mathbf{p}'_{ij}))) \tag{10}$$

As shown in Fig. 3, after sinusoidal encoding, the signal differences between $\mathbf{p}'_{ij}$ and $\mathbf{p}'_{ik}$ are amplified.

### 3.5 Theoretical Expressivity of PPGT

The theoretical expressivity of PPGT can be analyzed within the GD-WL framework (Zhang et al., 2023b). When equipped with regular generalized distances (e.g., RRWP, resistance distance (Zhang et al., 2023b)) as graph PE, PPGT attains expressivity strictly beyond 1-WL and is upper bounded by 3-WL, matching the theoretical limits of a broad class of GD-WL algorithms. We provide the formal proof in Appx. E.1.

## 4 Relationship with Previous Work

**Graph Transformers.** Transformers, especially *plain Transformers*—architectures close to the vanilla Transformer (Vaswani et al., 2017) with scaled dot-product (SDP) attention and feed-forward networks (FFNs)—have achieved outstanding performance across a wide range of domains, from language (Devlin et al., 2019; Raffel et al., 2020; OpenAI, 2024; Dubey et al., 2024) to vision (Dosovitskiy et al., 2021; Touvron et al., 2021b; Caron et al., 2021; Oquab et al., 2024).

Motivated by the success of Transformers in other domains, researchers have strived to migrate Transformers to graph learning to address the limitations of MPNNs. Although the naive migration of Transformers to graph learning did not work well (Dwivedi & Bresson, 2021), several recent works have achieved considerable success when constructing graph Transformers, from theoretical expressivity analysis (Zhang et al., 2023b; 2024b) to impressive empirical performance (Ying et al., 2021; Rampášek et al., 2022; Ma et al., 2023). ***However, during the development of these graph Transformers, there has been a gradual but substantial deviation from the plain Transformers widely used in other domains.*** For example, *SAN* (Kreuzer et al., 2021) introduces dual-attention mechanisms with local and global aggregations; *K-Subgraph SAT* (Chen et al., 2022) introduces MPNNs into attention mechanisms to compute attention scores; *GraphGPS* (Rampášek et al., 2022) heavily relies on the MPNNs within its hybrid Transformer architecture; *EGT* (Hussain et al., 2022) introduces gated mechanisms and edge-updates inside the attention; *GRIT* (Ma et al., 2023) incorporates a complicated conditional MLP-based attention mechanism (shown in

Appendix. D.1). These deviations prevent the easy adoption of Transformer training advances and obscure the potential unification of cross-modality foundation models. Among these graph Transformers, the *graphormer*-series (Ying et al., 2021; Luo et al., 2022; Zhang et al., 2023b) ***retain architectures that are closest to plain Transformers; unfortunately, the result is a substantial gap in empirical performance and empirical expressivity compared to the best-performing graph Transformers*** (Rampášek et al., 2022; Ma et al., 2023). Unlike most graph Transformers treating a node as a token, TokenGT (Kim et al., 2022) views both nodes and edges as tokens and processes them using plain or sparse Transformers. However, despite adopting a plain Transformer architecture, TokenGT also falls considerably behind recent graph Transformers on the expressivity-demanding PCQM4Mv2 dataset (as shown in Tab 4).[4]

Developing powerful graph Transformers based on a plain Transformer architecture is particularly attractive, as the associated hardware stacks and software libraries have already been extensively optimized. Therefore, we investigate the fundamental limitations of plain Transformers on graph-structured data and introduce several simple yet effective architectural enhancements that preserve the core plain Transformer design, enabling competitive empirical performance without requiring significant architectural modifications.

**Other Attention Mechanisms.** Besides SDP attention, there are other attention variants in use. Bahdanau et al. (2015) introduced the earliest content-based attention mechanism for recurrent neural networks (RNNs) based on an MLP, which is more computationally and memory costly. Swin-Transformer-V2 (Liu et al., 2022) uses cosine-similarity to compute attention scores for better stability, but neglects the magnitude information. Kim et al. (2021) propose the use of the negative of the square $L_2$-distance, with tied query-key projection weights, for maintaining Lipchitz continuity of Transformers. Our attention mechanism, although based on $L_2$ attention, is further simplified and adjusted in order to maintain alignment with SDP attention.

**Continuous Kernel Graph Convolution.** Ma et al. (2024) introduce graph convolution operators with continuous kernels defined over pseudo-coordinates of graphs, termed CKGConv. These operators offer better flexibility in capturing high-frequency information compared to attention-based mechanisms. PPGT with URPE enhancement can also be interpreted as a generalization of CKGConv. In Eq. (5), the $\phi(\mathbf{p}_{ij})$ term serves as the convolution kernel, analogous to that in CKGConv, while the $\texttt{Softmax}_j\left(\frac{\mathbf{q}_i^\intercal \mathbf{k}_j}{\sqrt{D}} - \frac{\mathbf{k}_j^\intercal \mathbf{k}_j}{2\sqrt{D}} + \theta(\mathbf{p}_{ij})\right)$ component can be regarded as a dynamic density function conditioned on the token representations, which is assumed to be uniform in CKGConv.

**Universality of Transformers** The Universal Approximation Theorem is fundamental for understanding the theoretical representational capacity upper bound of neural networks. While Hornik et al. (1989) established that MLPs are universal approximators for functions $f : \mathbb{R}^N \to \mathbb{R}^M$ on compact sets, this conclusion does not automatically extend to other domains. Yun et al. (2020) demonstrated that Transformers with absolute Positional Encodings (PE)—which sufficiently distinguish token order—are universal approximators for sequence-to-sequence functions. Conversely, Luo et al. (2022) showed that Transformers with relative PE lack this universality. However, they show that it can be restored through the universality enhancement. Regarding graph domains, Kreuzer et al. (2021) state that Transformers with absolute PE can approximate any function $f$ for the graph isomorphism problem. Crucially, however, this requires that the absolute PE can uniquely identify nodes in each graph, which is infeasible due to the highly symmetric structure of graphs.

Graph Transformers with full attention and appropriate positional encoding (PE) are typically considered at most as expressive as the 3-WL test for graph isomorphism (Zhang et al., 2023b). In contrast, those using linear attention have expressivity equivalent to MPNNs with virtual nodes (Cai et al., 2023).

**Additional related work.** We discuss additional related work, including MPNNs, graph positional/structural encoding, higher-order GNNs, and subgraph GNNs in Appx. D.

---

[4]We provide an in-depth comparison with TokenGT in Appx. D.6.

# 5 Experimental Results

## 5.1 Empirical Expressivity on Graph Isomorphism

To better understand the expressivity of PPGT, we evaluate our model on the BREC benchmark (Wang & Zhang, 2024), a comprehensive dataset for measuring the empirical expressive power of GNNs w.r.t. graph isomorphism, with graph-pairs from 1-WL to 4-WL-indistinguishable.

From the results (as shown in Tab. 1), we can uncover several conclusions and insights:

**[1]. GTs can reach empirical expressivity approaching the theoretical expressivity:** With proper architectural designs and graph PE, most GTs achieve decent expressivity bounded by 3-WL, matching 3-WL equivalence on the Basic, Regular, and Extended categories of graph pairs. Graphormer, as an example of earlier plain Transformers, demonstrates inferior expressivity, *highlighting the necessity of improving plain Transformers for graph structure learning*. PPGT, while maintaining a plain Transformer architecture, achieves powerful empirical expressivity through our proposed modifications, surpassing other GTs with more sophisticated architectures.

**[2]. Mismatch between theoretical and empirical expressivity:** The theoretical expressivity is not completely reflected in the empirical expressivity. For example, despite the same theoretical expressivity, adding SPE—which enhances the information extraction from PE—to PPGT can significantly boost the empirical expressivity, distinguishing 24 pairs of graphs in CFI (improved from 8 pairs). It is worth noting that applying SPE to RRWP does not introduce additional structural information about the graph. Rather, it alleviates the spectral bias inherent in neural network optimization (Rahaman et al., 2019; Mildenhall et al., 2020).. On the other hand, EPNN and PPGN, despite having stronger theoretical expressivity, achieve worse empirical expressivity compared to PPGT. This indicates that *besides theoretical expressivity, whether GNNs can effectively learn to fulfill their theoretical expressivity also matters*.

**[3]. Mismatch between expressivity and real-world benchmark performance:** The stronger theoretical/empirical expressivity is not completely reflected in real-world benchmark performance. For example, subgraph GNNs and/or K-WL GNNs with stronger expressivity (e.g., SSWL+, $I^2$GNN, $N^2$GNN) demonstrate inferior performance compared to GRIT and PPGT on the ZINC benchmark.

**[4]. Going beyond GD-WL?:** As previously discussed, the design of graph PE, a.k.a., pseudo-coordinates, can be extended beyond the distance/affinity of graphs. With this in mind, we conduct an exploratory demo called $I^2$GNN+PPGT, which uses $I^2$GNN to generate additional positional encodings for PPGT. The empirical expressivity is further improved to 76%, outperforming the standalone $I^2$GNN and PPGT, surpassing 3-WL and reaching the top performance among the methods compared. This demonstration hints that *plain Graph Transformers can potentially surpass GD-WL and achieve greater expressive power purely through enhanced positional encoding designs*. This result is noteworthy, as PPGT achieves superior empirical performance compared to many subgraph GNNs and higher-order GNNs that possess greater theoretical expressivity.

**The expressivity bottleneck of our PPGT model does not stem from its architecture, but rather from the design of positional encodings (PE). To fully realize the expressive potential of plain Graph Transformers, it is crucial to develop expressive and generalizable PE schemes for graphs that are also compatible with permutation symmetry.**

## 5.2 Benchmarking PPGT on Real-world Benchmarks

The specifications, details, and references for the baseline methods are provided in Appx. A.6.

**Benchmarking GNNs** We conduct a general evaluation of our proposed PPGT on five datasets from *Benchmarking GNNs* (Dwivedi et al., 2022a): ZINC, MNIST, CIFAR10, PATTERN, and CLUSTER, and summarize the results in Table 2. We observe that our model obtains the best mean performance for all five datasets, outperforming various MPNNs, non-MPNN GNNs, and existing graph Transformers. These results showcase the effectiveness of PPGT for general graph learning with a plain Transformer architecture.

**Long Range Graph Benchmarks** We present experimental results on three *Long-range Graph Benchmark (LRGB)* (Dwivedi et al., 2022c) datasets – Peptides-Function, Peptides-Structure and PASCALVOC-SP

Table 1: Theoretical Expressivity (WL-Class) vs. Empirical Expressivity (*BREC*) vs. Empirical Performance (*ZINC-12K*). Notations on expressivity follow previous works (Morris et al., 2020; Zhang et al., 2024b) that $\equiv$: equivalent; A $\sqsupset$ ($\sqsupseteq$) B: A is bounded by (or equivalent to) B; A $\not\sqsupseteq$ B: A is not bounded by B. $\cdot$ for unnamed WL-class. $(k-1)$-FWL $\equiv$ $k$-WL, for $k > 2$.

| Type | Model | WL-Class | Basic (60) Num.(↑) | Reg.(140) Num.(↑) | Ext. (100) Num.(↑) | CFI (100) Num.(↑) | Total (400) Num.(↑) | Total (400) Acc. (↑) | ZINC MAE (↓) |
|---|---|---|---|---|---|---|---|---|---|
| Heuristic Algorithm | 1-WL | 1-WL | 0 | 0 | 0 | 0 | 0 | 0% | - |
| | 3-WL | 3-WL | 60 | 50 | 100 | 60 | 270 | 67.5% | - |
| Subgraph GNNs | SUN | SWL $\sqsupset$ 3-WL | 60 | 50 | 100 | 13 | 223 | 55.8% | 0.083 |
| | SSWL+ | SWL $\sqsupset$ SSWL $\sqsupset$ 3-WL | 60 | 50 | 100 | 38 | 248 | 62% | 0.070 |
| | I$^2$GNN | $\cdot : \not\sqsupseteq$ 3-WL | 60 | 100 | 100 | 21 | 281 | 70.2% | 0.083 |
| K-WL GNNs | PPGN | 3-WL | 60 | 50 | 100 | 23 | 233 | 58.2% | - |
| | 2-DRFWL(2) | $\cdot$ $\sqsupset$ 2-FWL | 60 | 50 | 99 | 0 | 209 | 52.25 % | 0.077 |
| | 3-DRFWL(2) | $\cdot$ $\sqsupset$ 2-FWL | 60 | 50 | 100 | 13 | 223 | 55.75 % | - |
| | N$^2$GNN | 2-FWL $\sqsupseteq$ 2-FWL+ $\sqsupset$ 3-FWL | 60 | 100 | 100 | 27 | 287 | 71.8% | 0.059 |
| GD-WL GNNs | Graphormer | GD-WL $\sqsupset$ 3-WL | 16 | 12 | 41 | 10 | 79 | 19.8% | 0.122 |
| | EPNN | GD-WL $\sqsupseteq$ EPWL $\sqsupset$ 3-WL | 60 | 50 | 100 | 5 | 215 | 53.8% | - |
| | CKGConv | GD-WL $\sqsupset$ 3-WL | 60 | 50 | 100 | 8 | 218 | 54.5% | 0.059 |
| | GRIT | GD-WL $\sqsupset$ 3-WL | 60 | 50 | 100 | 8 | 218 | 54.5% | 0.059 |
| | PPGT w/o SPE | GD-WL $\sqsupset$ 3-WL | 60 | 50 | 100 | 8 | 234 | 54.5% | - |
| | PPGT | GD-WL $\sqsupset$ 3-WL | 60 | 50 | 100 | 24 | 234 | 58.5% | 0.057 |
| | I$^2$GNN+PPGT | GD++-WL | 60 | 120 | 100 | 24 | 304 | 76% | - |

Table 2: Test performance on five benchmarks from *Benchmarking GNNs* (baselines please see Appx. A.6 for details and references). Shown is the mean $\pm$ s.d. of 4 runs with different random seeds. Highlighted are the top first, second, and third results. # Param under $500K$ for ZINC, PATTERN, CLUSTER and $\sim 100K$ for MNIST and CIFAR10.

| Model | ZINC MAE↓ | MNIST Accuracy↑ | CIFAR10 Accuracy↑ | PATTERN W. Accuracy↑ | CLUSTER W. Accuracy↑ |
|---|---|---|---|---|---|
| | | Message Passing Networks | | | |
| GCN | $0.367 \pm 0.011$ | $90.705 \pm 0.218$ | $55.710 \pm 0.381$ | $71.892 \pm 0.334$ | $68.498 \pm 0.976$ |
| GIN | $0.526 \pm 0.051$ | $96.485 \pm 0.252$ | $55.255 \pm 1.527$ | $85.387 \pm 0.136$ | $64.716 \pm 1.553$ |
| GAT | $0.384 \pm 0.007$ | $95.535 \pm 0.205$ | $64.223 \pm 0.455$ | $78.271 \pm 0.186$ | $70.587 \pm 0.447$ |
| GatedGCN | $0.282 \pm 0.015$ | $97.340 \pm 0.143$ | $67.312 \pm 0.311$ | $85.568 \pm 0.088$ | $73.840 \pm 0.326$ |
| PNA | $0.188 \pm 0.004$ | $97.94 \pm 0.12$ | $70.35 \pm 0.63$ | $-$ | $-$ |
| | | Non-MPNN Graph Neural Networks | | | |
| CRaW1 | $0.085 \pm 0.004$ | $97.944 \pm 0.050$ | $69.013 \pm 0.259$ | $-$ | $-$ |
| GIN-AK+ | $0.080 \pm 0.001$ | $-$ | $72.19 \pm 0.13$ | $86.850 \pm 0.057$ | $-$ |
| DGN | $0.168 \pm 0.003$ | $-$ | $72.838 \pm 0.417$ | $86.680 \pm 0.034$ | $-$ |
| CKGCN | $0.059 \pm 0.003$ | $98.423 \pm 0.155$ | $72.785 \pm 0.436$ | $88.661 \pm 0.143$ | $79.003 \pm 0.140$ |
| | | Graph Transformers | | | |
| SAN | $0.139 \pm 0.006$ | $-$ | $-$ | $86.581 \pm 0.037$ | $76.691 \pm 0.65$ |
| K-Subgraph SAT | $0.094 \pm 0.008$ | $-$ | $-$ | $86.848 \pm 0.037$ | $77.856 \pm 0.104$ |
| EGT | $0.108 \pm 0.009$ | $98.173 \pm 0.087$ | $68.702 \pm 0.409$ | $86.821 \pm 0.020$ | $79.232 \pm 0.348$ |
| Graphormer-GD | $0.081 \pm 0.009$ | $-$ | $-$ | $-$ | $-$ |
| GPS | $0.070 \pm 0.004$ | $98.051 \pm 0.126$ | $72.298 \pm 0.356$ | $86.685 \pm 0.059$ | $78.016 \pm 0.180$ |
| GMLP-Mixer | $0.077 \pm 0.003$ | $-$ | $-$ | $-$ | $-$ |
| GRIT | $0.059 \pm 0.002$ | $98.108 \pm 0.111$ | $76.468 \pm 0.881$ | $87.196 \pm 0.076$ | $80.026 \pm 0.277$ |
| PPGT | $0.0566 \pm 0.002$ | $98.614 \pm 0.096$ | $78.560 \pm 0.700$ | $89.752 \pm 0.030$ | $80.027 \pm 0.114$ |

in Table 3. PPGT achieves the lowest MAE on Peptides-Structure, top F1 on PascalVoc-SP and remains in the top three models on Peptides-Function. We adopt the updated experimental setup from LRGB as described in Tönshoff et al. (2024), and report the corresponding results. Moreover, the difference in AP for Peptides-Function dataset between the best performing GRIT and our PPGT is not statistically significant

Table 3: Performance comparison on *Long-Range Graph Benchmark (LRGB) – Peptides* and *PascalVoc-SP* datasets. (mean ± s.d. of 4 runs). Highlighted are the top first, second, and third results.

| Method | Peptides-Func | Peptides-Struct | PascalVoc-SP |
|---|---|---|---|
| | **AP ↑** | **MAE ↓** | **F1 ↑** |
| GCN | $0.6860 \pm 0.0050$ | $0.2460 \pm 0.0007$ | $0.2078 \pm 0.0031$ |
| GINE | $0.6621 \pm 0.0067$ | $0.2473 \pm 0.0017$ | $0.2718 \pm 0.0054$ |
| GatedGCN | $0.6765 \pm 0.0047$ | $0.2477 \pm 0.0009$ | $0.3880 \pm 0.0040$ |
| DRew | $0.7150 \pm 0.0044$ | $0.2536 \pm 0.0015$ | $0.3314 \pm 0.0024$ |
| Exphormer | $0.6527 \pm 0.0043$ | $0.2481 \pm 0.0007$ | $0.3960 \pm 0.0027$ |
| GPS | $0.6534 \pm 0.0090$ | $0.2509 \pm 0.0010$ | $0.4440 \pm 0.0065$ |
| GRIT | $0.6988 \pm 0.0082$ | $0.2460 \pm 0.0012$ | – |
| PPGT | $0.6961 \pm 0.0062$ | $0.2450 \pm 0.0017$ | $0.4641 \pm 0.0033$ |

at the 5% level for a one-tailed t-test. These results demonstrate PPGT's capability of learning long-range dependency structures.

### 5.3 Benchmarking PPGT on Large-scale Graph Benchmark and Large-scale-graph Benchmark

Scaling behavior with respect to both model size and dataset size is a critical consideration. Accordingly, in this section, we evaluate the applicability of PPGT to large-scale data settings without imposing a strict parameter budget.

We consider two distinct types of large-scale data settings of graphs: *large-scale graph benchmarks* and *large-scale-graph benchmarks*. Although prior work often conflates these two scenarios, the distinction is important.

Specifically, *large-scale (graph) datasets* comprise many graph instances, requiring scalability with respect to dataset size. In contrast, *large-scale graph datasets* consist of a single graph with an extremely large number of nodes, necessitating scalability with respect to input size (analogous to Gigapixel image processing). A more detailed discussion is provided in Appendix F.1.

**Large-scale Graph Benchmark: PCQM4Mv2**  To further assess the scalability of PPGT to large-scale data, we evaluate its performance on the PCQM4Mv2 large-scale graph regression benchmark, which consists of 3.7 million graphs (Hu et al., 2021). This dataset is among the largest-scale graph benchmarks to date (see Table 4). Following the experimental protocol of Rampášek et al. (2022), we exclude the 3D information from the model attributes and use the PCQM4Mv2 validation set in place of the original private *Test-dev* set for evaluation. By omitting 3D information, we aim to more accurately measure each model's intrinsic graph learning capability, ensuring that results do not reflect reliance on 3D coordinates.

The results are reported from a single random seed due to the large scale of the dataset, following prior work.

We report the performance of PPGT-small (17.6M parameters), which is comparable in size to GRIT, and PPGT-base (86.8M parameters), which is comparable to ViT-Base (Dosovitskiy et al., 2021).

PPGT-small achieves slightly better performance than GRIT and GraphGPS (GPS-medium) under a comparable parameter budget. PPGT-base further improves performance, indicating that scaling PPGT yields consistent gains, even without increasing the dataset size.

**Large-scale-graph Benchmark: OGBN-ArXiv**  Unlike large-scale graph benchmarks, which prioritize the model's capacity to learn from many graph instances, a large-scale-graph benchmark typically emphasizes memory efficiency to handle the substantial size of an individual graph.

Even though PPGT is inherently not designed for processing large-size inputs, we also benchmark our approach on a large-scale-graph dataset: OGBN-ArXiv (Hu et al., 2020)[5].

---

[5]No feature enhancement techniques, such as UniMP (Shi et al., 2021) or GIANT (Chien et al., 2022), are applied.

Table 4: Test performance on *PCQM4Mv2* dataset. Shown is the result of a single run due to the computation constraint. Highlighted are the top **first**, **second**, and **third** results.

| Method | Model | Valid. (MAE $\downarrow$) | # Param |
|---|---|---|---|
| MPNNs | GCN | 0.1379 | 2.0M |
| | GCN-virtual | 0.1153 | 4.9M |
| | GIN | 0.1195 | 3.8M |
| | GIN-virtual | 0.1083 | 6.7M |
| Graph Transformers | GRPE | 0.0890 | 46.2M |
| | Graphormer | 0.0864 | 48.3M |
| | TokenGT (ORF) | 0.0962 | 48.6M |
| | TokenGT (Lap) | 0.0910 | 48.5M |
| | GPS-small | 0.0938 | 6.2M |
| | GPS-medium | 0.0858 | 19.4M |
| | GRIT | 0.0859 | 16.6M |
| | PPGT-small (Ours) | 0.0856 | 17.6M |
| | PPGT-base (Ours) | 0.08396 | 86.8M |

To handle large graphs, we adopt an additional graph sampling strategy. Specifically, we convert node-level tasks into graph-level tasks by extracting a local induced subgraph around each target node using breadth-first search (BFS) node sampling, referred to as Node2Subgraph conversion. Implementation details of Node2Subgraph can be found in Appx. B.4 Similar techniques exist (Zeng et al., 2020; Sun et al., 2023), but we do not explore them here, as they are beyond the focus of this work. Even in efficiency-oriented graph Transformers, graph sampling techniques are widely adopted to handle ultra-large-scale input graphs that cannot be processed directly.

We compare PPGT with several state-of-the-art efficiency-oriented graph Transformers, including Node-Former (Wu et al., 2022), Exphormer (Shirzad et al., 2023), and SGFormer (Wu et al., 2023). These models aim for full node coverage on large graphs but often compromise expressivity, as linear-attention graph Transformers exhibit the equivalent theoretical expressivity as MPNNs with virtual nodes (Cai et al., 2023). PPGT maintains high expressivity through Node2Subgraph sampling, even at the cost of reduced node coverage, and achieves performance comparable to these SOTA efficiency-oriented graph Transformers.

These results (as shown in Tab. 5) suggest that both model expressivity and node coverage are essential for learning on large graphs, and achieving a balance between the two is more important than overemphasizing either.

Table 5: Testing results (Accuracy) on OGBN-ArXiv (mean $\pm$ s.d. of 4 runs). Highlighted are the top **first**, **second**, and **third** results. Baseline results from (Wu et al., 2023; Shirzad et al., 2023; Li et al., 2022). (*DeeperGCN does not provide s.d.)

| Method | GCN | GCN-NSampler | DeeperGCN | SIGN | NodeFormer | SGFormer | Exphormer | PPGT |
|---|---|---|---|---|---|---|---|---|
| **OGBN-ArXiv** | $71.74 \pm 0.29$ | $68.50 \pm 0.23$ | 71.90 | $70.28 \pm 0.25$ | $59.90 \pm 0.42$ | $72.63 \pm 0.13$ | $72.44 \pm 0.28$ | $72.46 \pm 0.15$ |

## 5.4 Ablation Study on Proposed Designs

We perform a detailed ablation experiment on ZINC to study the usefulness of each architectural modification proposed in this work. From Figure 4, we observe that replacing the complicated, conditional MLP-based attention computation in GRIT (Ma et al., 2023) by SDP attention leads to worse performance if BN (Ioffe & Szegedy, 2015) is used. This suggests that BN's inability to regulate the token magnitude information hurts performance. Using $sL_2$ attention with BN is slightly better, showing that the $sL_2$ attention improves over SDP by mitigating the bias towards large magnitude keys. The same trend holds for AdaRMSN as well. Moreover, SDP+ARMSN performs better than SDP+RMSN, showing that the flexibility of preserving the

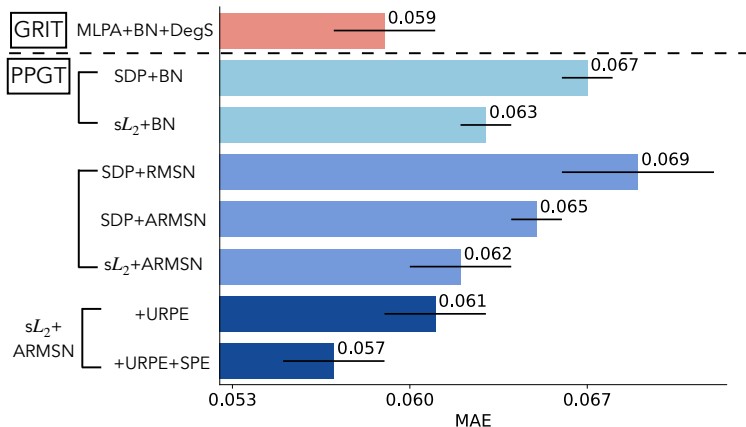

Figure 4: Ablation Study on *ZINC*. MLPA: Conditional MLP Attention; DegS: degree scaler; ARMSN: AdaRMSNorm; URP: Universal RPE; SPE: Sinusoidal PE enhancement.

magnitude information contributes positively towards performance. Finally, we observe that the use of URPE and sinusoidal PE enhancement provides additional benefits.

### 5.5 Further Study

In Appx. C, we present additional experimental analyses, including: (C.1) a sensitivity study on the number of bases $S$ in SPE, demonstrating the impact of SPE on empirical expressivity; (C.2) a case study of different normalization layers, showing that RMSN discards magnitude information whereas AdaRMSN preserves it; (C.3) a batch-size sensitivity analysis of AdaRMSN, demonstrating its robustness to batch size, in contrast to BN; and (C.6) a comparison of runtime and GPU memory consumption between PPGT and GRIT.

## 6 Conclusion

Plain Transformers are ill-suited for handling the unique challenges posed by graphs, such as the lack of canonical coordinates and permutation invariance. To obtain superior capacity and empirical performance, previous graph Transformers (GTs) have introduced non-standard, sophisticated, and domain-specific architectural modifications. In this work, we demonstrate that plain Transformers can be powerful graph learners via the proposed minimal, easy-to-adapt modifications. Our Powerful Plain Graph Transformers (PPGTs) not only achieve competitive expressivity, but also demonstrate strong empirical performance on real-world graph benchmarks while maintaining the simplicity of plain Transformer architectures. Further exploration of empirical expressivity also unveils a potential direction for improving plain GTs.

We consider this work an important first step toward reducing dissimilarities between GTs and Transformers in other domains. This may facilitate the design of general multimodal foundation models that integrate learning capabilities on graphs and potentially other irregular non-Euclidean geometric spaces.

Beyond graph domains, the insights from our work may also benefit Transformer architectures in other domains, broadening their applicability and impact.

**Limitations:** This work demonstrates that plain Transformers can serve as powerful graph learners without external graph models (e.g., MPNNs). However, like most vanilla Transformer architectures, PPGTs incur an $\mathcal{O}(N^2)$ computational complexity, which limits their direct applicability to large-size input graphs. To mitigate this issue, graph sampling techniques can be employed to handle large inputs more efficiently. A more detailed discussion of these limitations, along with potential directions for addressing them, is provided in Appx. F.

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

# A    Experimental Details

## A.1    Benchmarking GNNs and Long Range Graph Benchmarks

**Description of Datasets**    Table 6 provides a summary of the statistics and characteristics of the datasets used in this paper. The first five datasets are from Dwivedi et al. (2022a), and the last two are from Dwivedi et al. (2022c). Readers are referred to Rampášek et al. (2022) for more details of the datasets.

Table 6: Overview of the graph learning datasets involved in this work (Dwivedi et al., 2022a;c; Irwin et al., 2012; Hu et al., 2021).

| Dataset | # Graphs | Avg. # nodes | Avg. # edges | Directed | Prediction level | Prediction task | Metric |
|---|---|---|---|---|---|---|---|
| ZINC | 12,000 | 23.2 | 24.9 | No | Graph | Regression | Mean Abs. Error |
| SP-MNIST | 70,000 | 70.6 | 564.5 | Yes | Graph | 10-class classif. | Accuracy |
| SP-CIFAR10 | 60,000 | 117.6 | 941.1 | Yes | Graph | 10-class classif. | Accuracy |
| PATTERN | 14,000 | 118.9 | 3,039.3 | No | Inductive Node | Binary classif. | Weighted Accuracy |
| CLUSTER | 12,000 | 117.2 | 2,150.9 | No | Inductive Node | 6-class classif. | Weighted Accuracy |
| Peptides-func | 15,535 | 150.9 | 307.3 | No | Graph | 10-task classif. | Avg. Precision |
| Peptides-struct | 15,535 | 150.9 | 307.3 | No | Graph | 11-task regression | Mean Abs. Error |
| PascalVoc-SP | 11,355 | 479.4 | 2,710.5 | No | Inductive Node | 21-class classif. | Marco F1 |
| PCQM4Mv2 | 3,746,620 | 14.1 | 14.6 | No | Graph | regression | Mean Abs. Error |
| OGBN-ArXiv | 1 | 169,343 | 1,116,243 | Yes | Transductive Node | 40-class classif. | Accuracy |

**Dataset splits, random seed, and parameter budgets**    We conduct the experiments on the standard train/validation/test splits of the evaluated benchmarks, following previous works (Rampášek et al., 2022; Ma et al., 2023). For each dataset, we execute 4 trials with different random seeds (0, 1, 2, 3) and report the mean performance and standard deviation. We follow the most commonly used parameter budgets: around 500k parameters for ZINC, PATTERN, CLUSTER, Peptides-func, and Peptides-struct; and around 100k parameters for SP-MNIST and SP-CIFAR10.

**Hyperparameters**    Due to the limited time and computational resources, we did not perform an exhaustive search on the hyperparameters. We start with the hyperparameter setting of GRIT (Ma et al., 2023) and perform minimal search to satisfy the commonly used parameter budgets.

The hyperparameters are presented in Table 7 and Table 8. In the tables, S.D. stands for stochastic depth (Huang et al., 2016), a.k.a., drop-path, in which we treat a graph as one example in stochastic depth.

For Peptides-func, we find that there exists a mismatch between the cross-entropy loss and the metric, average precision, due to the highly imbalanced label distribution. Empirically, we observe that adding a BN in the prediction head post-graph-pooling effectively mitigates the problem.

## A.2    BREC: Empirical Expressivity Benchmark

BREC (Wang & Zhang, 2024) is an empirical expressivity dataset, consisting of four major categories of graphs: Basic, Regular, Extension, and CFI. *Basic graphs* include 60 pairs of simple 1-WL-indistinguishable graphs. *Regular graphs* include 140 pairs of regular graphs from four types of subcategories: 50 pairs of simple regular graphs, 50 pairs of strongly regular graphs, 20 pairs of 4-vertex condition graphs and 20 pairs of distance regular graphs. *Extension graphs* include 100 pairs of special graphs that arise when comparing four kinds of GNN extensions (Papp & Wattenhofer, 2022). *CFI graphs* include 100 pairs of graphs generated by CFI methods (Cai et al., 1992). All pairs of graphs are 1-WL-indistinguishable.

We follow the standard training pipelines from BREC: pairwise contrastive training and evaluating process. No specific parameter budget required on the BREC dataset.

**Hyperparameter**    The hyperparameter setting for PPGT on BREC can be found in Table 10.

Table 7: Hyperparameters for five datasets from Benchmarking GNNs (Dwivedi et al., 2022a)

| Hyperparameter | ZINC | MNIST | CIFAR10 | PATTERN | CLUSTER |
|---|---|---|---|---|---|
| **PE Stem** | | | | | |
| MLP-dim | 128 | 64 | 64 | 128 | 128 |
| Edge-dim | 64 | 32 | 32 | 64 | 64 |
| # FFN | 2 | 1 | 2 | 2 | 2 |
| FFN expansion | 2 | 2 | 2 | 2 | 2 |
| PE-dim | 24 | 10 | 10 | 32 | 32 |
| # bases | 3 | 3 | 3 | 3 | 3 |
| **Backbone** | | | | | |
| # blocks | 12 | 4 | 10 | 12 | 16 |
| Dim | 64 | 48 | 32 | 64 | 56 |
| FFN expansion | 2 | 2 | 2 | 2 | 2 |
| # attn. heads | 8 | 6 | 4 | 8 | 7 |
| S.D. | 0.1 | 0.2 | 0.2 | 0.1 | 0.3 |
| Attn. dropout | 0.2 | 0.5 | 0.2 | 0.1 | 0.3 |
| **Pred. Head** | | | | | |
| Graph Pooling | sum | mean | mean | - | - |
| Norm | - | - | - | - | - |
| # layers | 3 | 2 | 2 | 2 | 2 |
| **Training** | | | | | |
| Batch size | 32 | 32 | 16 | 32 | 16 |
| Learn. rate | 2e-3 | 1e-3 | 1e-3 | 1e-3 | 1e-3 |
| # epochs | 2500 | 400 | 200 | 400 | 150 |
| # warmup | 50 | 10 | 10 | 10 | 10 |
| Weight decay | 1e-5 | 1e-5 | 1e-5 | 1e-5 | 1e-5 |
| # parameters | 487K | 102K | 108K | 497K | 496K |

For the experimental setup of $I^2$-GNN + PPGT, we construct a subgraph for each edge within the 4-hop neighborhood and use RRWP as the node-attributes. We employ a 6-layer GIN with BN on each subgraph independently to get its representation. The subgraph representations are fed to PPGT as additional edge-attributes. The $I^2$-GNN and PPGT are trained end-to-end together.

### A.3 PCQM4Mv2 from OGB Large-Scale Challenge

PCQM4Mv2 is a large-scale quantum chemistry graph dataset benchmark, containing over 3.7M graphs, proposed from OGB Large-Scale Challenge (OGB-LSC) (Hu et al., 2021). The statistics of the dataset can be found in Table 6. The hyperparameter settings for PPGT-small and PPGT-base (the latter having a model size and configuration similar to ViT-Base (Dosovitskiy et al., 2021)) are provided in Table 9.

### A.4 OGBN-ArXiv from OGB Benchmark

OGBN-ArXiv is a large-scale graph benchmark featuring a transductive node classification task from Open Graph Benchmark (Hu et al., 2020). Unlike the other graph datasets used in this work, OGBN-ArXiv comprises a single large-scale graph containing over 169,000 nodes. This benchmark emphasizes the graph models' scalability to large input graphs. The statistics of the dataset can be found in Table 6.

For PPGT, we convert the transductive node-classification task on OGBN-ArXiv into a graph-classification setting via the Node2Subgraph transformation. In this framework, we employ the class-attention mechanism (Touvron et al., 2021b) as the output head. As OGBN-ArXiv is a directed graph, we compute

Table 8: Hyperparameters for two datasets from the Long-range Graph Benchmark (Dwivedi et al., 2022c)

| Hyperparameter | Peptides-func | Peptides-struct | PascalVoc-SP |
|---|---|---|---|
| **PE Stem** | | | |
| MLP-dim | 128 | 128 | 128 |
| Edge-dim | 64 | 64 | 64 |
| # FFN | 1 | 1 | 1 |
| FFN expansion | 2 | 2 | 2 |
| # bases | 3 | 3 | 3 |
| PE-dim | 32 | 24 | 16 |
| **Backbone** | | | |
| # blocks | 5 | 5 | 12 |
| Dim | 96 | 96 | 64 |
| FFN expansion | 2 | 2 | 2 |
| # attn. heads | 16 | 8 | 8 |
| S.D. | 0.1 | 0.1 | 0.2 |
| Attn. dropout | 0.2 | 0.1 | 0.3 |
| **Pred. Head** | | | |
| Graph Pooling | sum | mean | - |
| Norm | BN | - | - |
| # layers | 3 | 2 | 2 |
| **Training** | | | |
| Batch size | 32 | 32 | 16 |
| Learn. rate | 7e-4 | 7e-4 | 1e-3 |
| # epochs | 400 | 250 | 200 |
| # warmup | 10 | 10 | 10 |
| Weight decay | 1e-5 | 1e-5 | 1e-2 |
| # Parameters | 509K | 488K | 492K |

dual-direction RRWP features, i.e., random walks along both in-edge and out-edge directions. The corresponding hyperparameter setting is provided in Table 9.

### A.5 Optimizer and Learning Rate Scheduler

Following most plain Transformers in other domains, we use AdamW (Loshchilov & Hutter, 2019) as the optimizer and the Cosine Annealing Learning Rate scheduler (Loshchilov & Hutter, 2017) with linear warm up.

### A.6 Baselines Information

**For benchmarks from BenchmarkingGNN (Dwivedi et al., 2022a).**

- **Tyical Message-passing Networks(MPNNs):** GCN (Kipf & Welling, 2017), GIN (Xu et al., 2019), GAT (Veličković et al., 2018), GatedGCN (Bresson & Laurent, 2018), PNA (Corso et al., 2020);

- **GNNs going beyond MPNNs:** CRaW1 (Tönshoff et al., 2023), GIN-AK+ (Zhao et al., 2022), DGN (Beani et al., 2021), CKGCN (Ma et al., 2024),

Table 9: Hyperparameters for PCQM4Mv2 (Hu et al., 2021) and OGBN-ArXiv (Hu et al., 2020).

| Hyperparameter | PCQM4Mv2 (PPGT-small) | PCQM4Mv2 (PPGT-base) | OGBN-ArXiv |
|---|---|---|---|
| **PE Stem** | | | |
| MLP-dim | 512 | 512 | 192 |
| Edge-dim | 128 | 128 | 96 |
| # FFN | 8 | 2 | 2 |
| FFN expansion | 2 | 4 | 2 |
| # bases | 10 | 5 | 3 |
| PE-dim | 24 | 8 | 5 (dual direction) |
| **Backbone** | | | |
| # blocks | 16 | 12 | 10+2(class attention) |
| Dim | 512 | 768 | 192 |
| FFN expansion | 2 | 4 | 2 |
| # attn. heads | 16 | 12 | 8 |
| S.D. | 0.2 | 0.2 | 0.5 |
| Attn. dropout | 0.2 | 0.1 | 0.5 |
| **Node2Subgraph** | | | |
| Node Masking | - | - | 0.5 |
| Max Size of Graph | - | - | 100 |
| **Pred. Head** | | | |
| Graph Pooling | sum | mean | class attention |
| Norm | - | - | |
| # layers | 3 | 3 | 1 |
| **Training** | | | |
| Batch size | 2048 | $2048 \times 2$GPU | 128 |
| Learn. rate | 1e-3 | 2e-3 | |
| # epochs | 500 | 300 | |
| # warmup | 50 | 20 | 10 |
| Weight decay | 1e-3 | 7e-4 | 5e-2 |
| # parameters | 17.6M | 86.8M | 3.74M |

- **Graph Transformers** SAN (Kreuzer et al., 2021), K-Subgraph SAT (Chen et al., 2022), EGT (Hussain et al., 2022), Graphormer-GD (Zhang et al., 2023b), GPS (Rampášek et al., 2022), GMLP-Mixer (He et al., 2023), GRIT (Ma et al., 2023).

**For BREC (Wang & Zhang, 2024):**

- **Subgraph GNNs**: SUN (Frasca et al., 2022), SSWL+ (Zhang et al., 2023a), $I^2$-GNN (Huang et al., 2023)

- **K-WL/K-FWL GNNs**: PPGN (Maron et al., 2019), 2-DRFWL(2) (Zhou et al., 2023), 3-DRFWL(2) (Zhou et al., 2023)

- **K-FWL+SubgraphGNNs**: $N^2$GNN (Feng et al., 2023)

- **GD-WL GNNs** Graphormer (Ying et al., 2021), EPNN (Zhang et al., 2024b), CKGConv (Ma et al., 2024), GRIT (Ma et al., 2023).

Table 10: Hyperparameters for BREC (Wang & Zhang, 2024)

| Hyperparameter | BREC | |
|---|---|---|
| **PE Stem** | | |
| MLP-dim | 192 | |
| Edge-dim | 96 | |
| # FFN | 4 | |
| FFN expansion | 2 | |
| # bases | 15 | |
| **Backbone** | | |
| # blocks | 6 | |
| Dim | 96 | |
| FFN expansion | 2 | |
| # attn. heads | 16 | |
| S.D. | 0. | |
| Attn. dropout | 0. | |
| **Pred. Head** | | |
| Graph Pooling | sum | |
| Norm | BN | - |
| # layers | 3 | |
| PE-dim | 32 | |
| **Training** | | |
| Batch size | 32 | |
| Learn. rate | 1e-3 | |
| # epochs | 200 | |
| # warmup | 10 | |
| Weight decay | 1e-5 | |
| # parameters | 874K | |

**For OGB-LSC (Hu et al., 2021):**
MPNNs (GCN (Kipf & Welling, 2017), GIN (Xu et al., 2019) with/without virtual nodes) as well as several Graph Transformers (GRPE (Park et al., 2022), Graphormer (Ying et al., 2021), TokenGT (Kim et al., 2022) and GraphGPS (Rampášek et al., 2022)).

## B    Implementation Details

### B.1    Attention in Graph Transformers

In SAN (Kreuzer et al., 2021) and GRIT (Ma et al., 2023), the attention mechanism is implemented with the sparse operations in *PyTorch Geometric* due to the complicated attention mechanism. For PPGTs, we provide two versions, one based on the sparse operations and another one based on the dense operations with padding. The latter one is typically faster for larger-scale graphs but consumes more memory.

### B.2    Initializations of Parameters

Following the most recent plain Transformers, we initialize the weights of linear layers in the backbone as well as the prediction heads with truncated normalization with standard deviation $\sigma = 0.02$.

For the stems, we utilize Kaiming uniform initialization (He et al., 2015) with $a = 0$ for hidden layers in MLPs and $a = 1$ for output layers in MLPs or standalone linear layers.

For lookup-table-like embedding layer, *nn.Embedding*, we utilize the default normal initialization with $\sigma = 1$.

### B.3   Injection of Degree and Graph Order

For all datasets, we inject the log-degree of each node and the log-graph-order (i.e., the number of nodes in the graph) as additional node attributes. Besides RRWP, we inject the reciprocal of degrees for node $i$ and node $j$ as well as the reciprocal of the graph order to $\mathbf{p}'_{ij}$ as an extension.

For superpixel datasets (CIFAR10, MNIST), we also include the location of pixel into the graph PE.

### B.4   Node2Subgraph

For large-scale graphs, we adopt a simple graph sampling strategy, termed *Node2Subgraph*, for PPGT.

We emphasize that this is a rudimentary sampling scheme, primarily included for benchmarking node-level tasks on large-scale graphs. More sophisticated sampling strategies could further enhance the performance of PPGT.

Specifically, we transform node-level tasks into graph-level tasks by extracting a local induced subgraph around each target node via breadth-first search (BFS). For each target node, we construct an induced subgraph up to $K$-hop, subject to a maximum size constraint of $M$ nodes. The extraction is done via BFS, i.e., $k$-hop from 1 to $K$. If the resulting $k$-hop subgraph exceeds $M$, we instead revert to the $(k{-}1)$-hop subgraph and supplement it by randomly sampling nodes from the $k$-hop frontier (i.e., nodes at shortest-path distance $k$ but not included in the $(k{-}1)$-hop subgraph) until the subgraph reaches $M$ nodes. This procedure yields an induced subgraph of fixed size $M$ while preserving connectivity around the target node.

We provid an extra sensitivity study on Node2Subgraph on Appx. C.5.

### B.5   Notes on Reproducibility

Our implementation is built upon *PyTorch* and *PyTorch Geometric*. For processing graphs, we utilize the *scatter* operations from *PyTorch Geometric*, which are known to be non-deterministic for execution on GPUs. Therefore, even with the same random seed, the experimental results of different trials might vary in an acceptable range. This statement is applicable to most existing models.

We conducted the experiments for ZINC with *NVIDIA GeForce RTX 4080 super*, experiments for PCQM4Mv2 on *NVIDIA H100*, and the rest experiments with *NVIDIA A100*.

## C   Additional Study

### C.1   Sensitivity Study: Impact of the Number of Bases $S$ in SPE on Empirical Expressivity

We conduct a sensitivity analysis on the hyperparameter $S$ in the sinusoidal PE enhancement, focusing on its impact on empirical expressivity. We notice that $S$ has no impact on distinguishing basic, regular, and extension graphs, but it displays a stronger influence on distinguishing CFI graphs. In this sensitivity study, we fix the other architectural components and hyperparameters and only vary $S$ along with the size of the first fully-connected layer.

As shown in Table 11, gradually increasing $S$ leads to a better ability to distinguish CFI graphs. This observation matches our motivation for incorporating SPE, as discussed in Sec. 3.4. SPE can effectively enhance the signal differences in graph PE, and thus can make it easier for MLPs to learn how to extract pertinent information. However, the accuracy does not increase monotonically with $S$. Once $S$ is sufficiently large, further increasing it does not necessarily lead to stronger empirical expressivity, and potentially can lead to overfitting and/or demand more training iterations.

Beyond empirical expressivity, we further conduct a sensitivity study on the real-world dataset *ZINC* (Table 12).

Table 11: Sensitivity study on $S$ in Sinusoidal encoding for RRWP on CFI graphs from BREC (Wang & Zhang, 2024).

| **SPE** ($S =$) | **0** | **3** | **6** | **9** | **12** | **15** | **20** |
|---|---|---|---|---|---|---|---|
| # Correct in CFI (100) | 3 | 3 | 9 | 17 | 20 | 24 | 22 |

A similar trend to that observed on CFI graphs emerges: incorporating SPE with a moderate number of bases $S$ improves performance, whereas overly large $S$ can degrade performance due to amplified high-frequency signals and increased risk of overfitting.

In this study, we fix all other hyperparameters, including regularization (e.g., dropout/drop-path rates and weight decay). Stronger regularization may help mitigate the adverse effects of large $S$, potentially leading to improved performance in that regime.

Table 12: Sensitivity study on $S$ in Sinusoidal encoding for RRWP on ZINC (Dwivedi et al., 2022a).

| **SPE** ($S =$) | **0** | **3** | **5** |
|---|---|---|---|
| ZINC (MAE ↓) | 0.0579± 0.0016 | 0.0566± 0.002 | 0.0587± 0.0018 |

## C.2 Case Study Comparing BN, RMSN and AdaRMSN

We conduct an additional case study to demonstrate the advantages of AdaRMSN in addressing the limitations of token-wise normalization: the ineffectiveness in preserving magnitude information.

We randomly generate a set of points in 2D Euclidean space with magnitudes ranging from 0.5 to 1.5. Given the data points, we train auto-encoders with BN/RMSN/AdaRMSN, respectively,

$$\mathbf{y} = \texttt{FC} \circ \texttt{Norm} \circ \texttt{FC}(\mathbf{x}) \tag{11}$$

where `FC` stands for fully-connected layers (i.e., linear layers) and `Norm` denotes the normalization layers. We conduct an overfitting test as a sanity check to measure information loss in the normalization layer via the autoencoders' ability to recover input data points in predictions. Each auto-encoder is trained independently for 5000 epochs in full-batch mode using the AdamW optimizer. As illustrated in Fig. 5, we present visualizations comparing input data points with predictions generated by autoencoders employing BN, RMSN, and AdaRMSN, respectively. The results demonstrate that RMSN fails to preserve magnitude information, while both BN and AdaRMSN successfully maintain this crucial aspect of the data points. We note that BN relies on cumulative moving averages of mini-batch statistics to estimate population parameters, which can make it sensitive to the choices of batch size.

## C.3 Sensitivity Study of AdaRMSN w.r.t. Batch Sizes

To better understand the advantages of AdaRMSN over BatchNorm (BN), we conducted a sensitivity study comparing normalization techniques w.r.t. batch sizes, based on PPGT on the ZINC dataset The study (as shown in Fig. 6) showcases the better stability of AdaRMSN compared to the choice of BN used in many previous GTs (Kreuzer et al., 2021; Rampášek et al., 2022; Ma et al., 2023).

## C.4 Stress Test of AdaRMSN on Initialization w.r.t. Learning Rates

In Sec. 3.1, we introduce the initialization of AdaRMSN. The key idea is to ensure that AdaRMSN behaves identically to standard RMSN at the initial stage, thereby guaranteeing training stability as RMSN.

To validate this design, we conduct a stress test (Table 13) under large learning rates, monitoring training over the first 50 epochs. We compare AdaRMSN with RMSN and a deliberately poor initialization (i.e., $\alpha = \mathbf{1}, \beta = \mathbf{0}$, corresponding to an identity mapping).

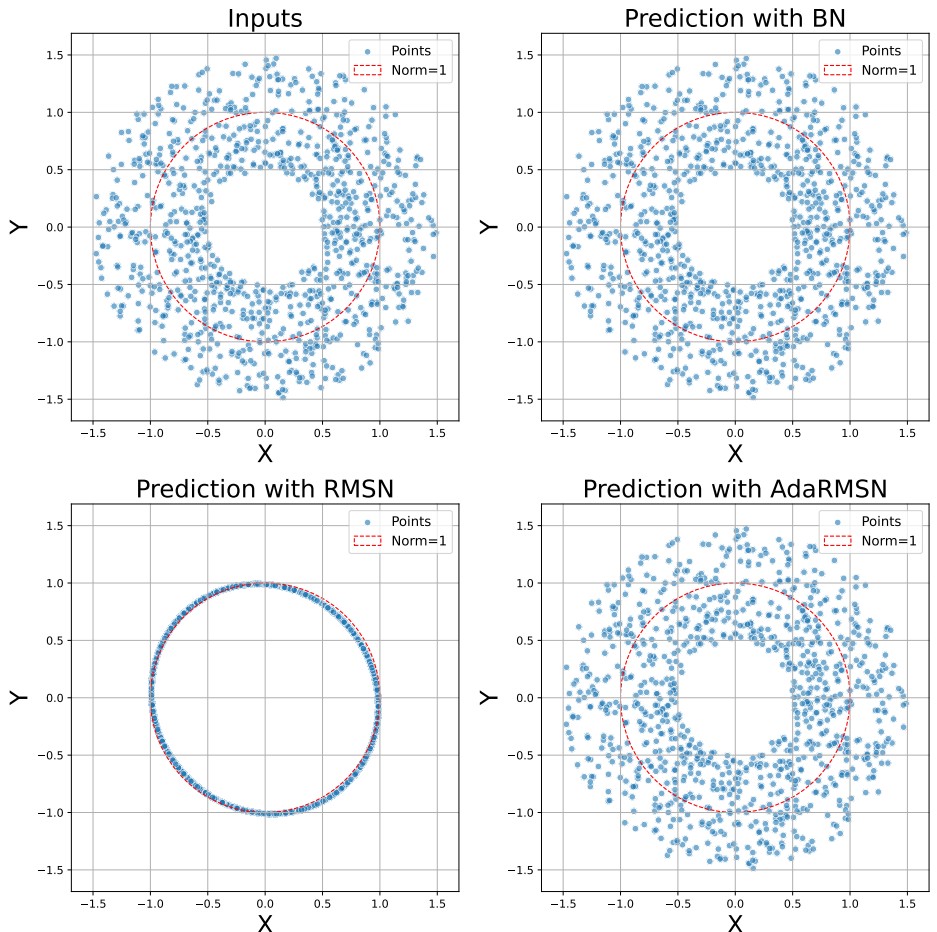

Figure 5: (Case Study of AdaRMSN) Visualization of Input and Pred data points [(1) Input; (2) Predictions w/ BN; (3) Predictions w/ RMSN; (4) Predictions w/ AdaRMSN]. RMSN is ineffective in preserving magnitude information, whereas both BN and AdaRMSN successfully maintain the crucial magnitude information of the data point

The results show that AdaRMSN domonstrates near-parity with RMSN on the stability, whereas the poorly initialized variant exhibits higher sensitivity to learning rates. This confirms that the proposed initialization is meaningful for maintaining stable training dynamics.

Table 13: Stress Test of AdaRMSN on Different Learning Rates. Conducted on ZINC dataset within first 50 epochs. 3 denotes the regular training while 7 denotes training crash (NaN or loss explosion). Bad-AdaRMSN refers to the variant initialized with $\alpha = \mathbf{1}$ and $\beta = \mathbf{0}$.

| Learning Rate | RMSN | AdaRMSN | Bad-AdaRMSN |
|:---:|:---:|:---:|:---:|
| 2e-3 | ✓ | ✓ | ✓ |
| 7e-3 | ✓ | ✓ | ✗ |
| 1e-2 | ✗ | ✗ | ✗ |
| 7e-2 | ✗ | ✗ | ✗ |

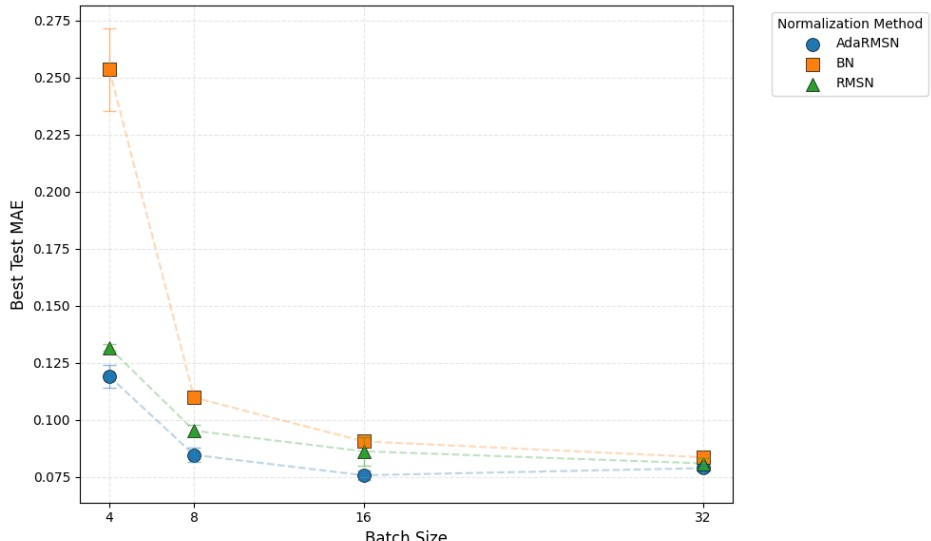

Figure 6: Test MAE of PPGT on ZINC v.s. Batch Size (BS). # Training epochs are adjusted per batch-size for the same total update steps: $400 * BS/32$. The first $10\%$ epochs are in the warmup stage. AdaRMSN and RMSN demonstrate better stability and less sensitivity to varying batch sizes compared to BN.

### C.5 Sensitivity Study on Node2Subgraph

In the main experiments of OGBN-ArXiv, we consider subgraphs up to $K = 15$ hops, with a maximum of $M = 100$ nodes per subgraph. Note that the effective $k$-hop for each induced subgraph may vary due to the rollback mechanism, and depends on the sparsity of the local structure.

We further conduct a sensitivity study on the choice of $K$, as reported in Table 14. The results show that once $K$ reaches a moderate value, the performance stabilizes. The optimal choice of $K$ may still vary across datasets.

Table 14: Sensitivity study on $K$-hop subgraphs in Node2Subgraph. The maximum number of nodes in each subgraph is set to 100. 0-hop denotes the setting where only target nodes are considered, without incorporating graph structure. Note that the effective $k$-hop for each induced subgraph may vary due to the rollback mechanism, and depends on the sparsity of the local structure.

| $K$-hop Subgraph | 0 | 1 | 3 | 5 | 10 | 15 |
|---|---|---|---|---|---|---|
| OGBN-ArXiv | 54.46 | 70.46 | 72.10 | 72.14 | 72.32 | 72.46 |

### C.6 Runtime and GPU Consumption

We report basic statistics on the runtime and GPU memory consumption of PPGT, in comparison with GRIT (see Table 15). The results suggest that, even without efficiency-driven techniques, adopting a plain Transformer architecture—rather than more complex graph-specific Transformer designs used in prior work—can yield improved runtime and reduced GPU memory consumption.

### C.7 Complexity of PPGT and Other Graph Transformers

We report the computational complexity of PPGT and other representative graph Transformers in Table 16. PPGT has the same asymptotic time and memory complexity as standard graph Transformers, i.e., $\mathcal{O}(N^2)$ per layer. GRIT employs an MLP-based attention mechanism, and GraphGPS incorporates an additional

Table 15: Comparison of peak GPU memory usage and per-epoch training time for GRIT and PPGT. Dataset: Peptides-Structure (15K graphs); Model config.: 5 transformer layers, 96 channels, batch size 32. Hardware: a single Nvidia V100 GPU with 32GB memory, supported by 80 Intel Xeon Gold 6140 CPUs running at 2.30GHz

| Model | GPU Memory (GB) | Training Time (Sec/Epoch) |
|---|---|---|
| GRIT | 29.16 | 141.60 |
| PPGT | 25.07 | 100.68 |
| Improv. | $\sim$14.03% | $\sim$28.9% |

MPNN branch; both introduce extra lower-order computational overhead. TokenGT treats edges as tokens, resulting in higher time and memory complexity, scaling as $D^2$, where $D$ denotes the average node degree ($D > 1$).

Table 16: Complexity comparison of representative graph Transformers. Here, $N$ denotes the number of nodes, $D$ denotes the average of node degree ($ND$ equals the number of edges). We report the dominant per-layer complexity with respect to the graph size, omitting hidden dimensions and constant factors.

| Model | Time Complexity (per layer) | Memory Complexity (per layer) |
|---|---|---|
| PPGT | $\mathcal{O}(N^2)$ | $\mathcal{O}(N^2)$ |
| GRIT | $\mathcal{O}(N^2)$ | $\mathcal{O}(N^2)$ |
| GraphGPS | $\mathcal{O}(N^2)$ | $\mathcal{O}(N^2)$ |
| Graphormer | $\mathcal{O}(N^2)$ | $\mathcal{O}(N^2)$ |
| TokenGT | $\mathcal{O}(N^2D^2)$ | $\mathcal{O}(N^2D^2)$ |

# D  Additional Related Work

## D.1  GRIT's Attention

The attention mechanism in GRIT (Ma et al., 2023) adopts the conditional MLP (Perez et al., 2018), involves linear projections, elementwise multiplications, and an uncommon non-linearity in the form of a signed-square-root:

$$
\begin{aligned}
\hat{\mathbf{e}}_{i,j} &= \sigma\Big(\rho\left((\mathbf{W}_{\mathrm{Q}}\mathbf{x}_i + \mathbf{W}_{\mathrm{K}}\mathbf{x}_j) \odot \mathbf{W}_{\mathrm{Ew}}\mathbf{e}_{i,j}\right) + \mathbf{W}_{\mathrm{Eb}}\mathbf{e}_{i,j}\Big) \in \mathbb{R}^{d'}, \\
\alpha_{ij} &= \mathrm{Softmax}_{j\in\mathcal{V}}(\mathbf{W}_{\mathrm{A}}\hat{\mathbf{e}}_{i,j}) \in \mathbb{R}, \\
\hat{\mathbf{x}}_i &= \sum_{j\in\mathcal{V}} \alpha_{ij} \cdot (\mathbf{W}_{\mathrm{V}}\mathbf{x}_j + \mathbf{W}_{\mathrm{Ev}}\hat{\mathbf{e}}_{i,j}) \in \mathbb{R}^d,
\end{aligned}
\tag{12}
$$

where $\sigma$ is a non-linear activation (ReLU by default); $\mathbf{W}_{\mathrm{Q}}, \mathbf{W}_{\mathrm{K}}, \mathbf{W}_{\mathrm{Ew}}, \mathbf{W}_{\mathrm{Eb}} \in \mathbb{R}^{d'\times d}$, $\mathbf{W}_{\mathrm{A}} \in \mathbb{R}^{1\times d'}$, $\mathbf{W}_{\mathrm{V}} \in \mathbb{R}^{d\times d}$ and $\mathbf{W}_{\mathrm{Ev}} \in \mathbb{R}^{d\times d'}$ are learnable weight matrices; $\odot$ indicates elementwise multiplication; and $\rho(\mathbf{x}) := (\mathrm{ReLU}(\mathbf{x}))^{1/2} - (\mathrm{ReLU}(-\mathbf{x}))^{1/2}$ is the signed-square-root. Note that $\mathbf{e}_{i,j}$ here corresponds to $\mathbf{p}_{ij}$ in our PPGT, but it requires updating within the attention mechanism. According to Ma et al. (2023), the signed-square-root $\rho$ is necessary to stabilize the training.

## D.2  MPNNs

As the most widely used GNNs, message-passing neural networks (MPNNs) (Gilmer et al., 2017; Kipf & Welling, 2017; Hamilton et al., 2017; Veličković et al., 2018) learn graphs, following the 1-WL framework (Xu et al., 2019). However, there are several known limitations of MPNNs: (1) over-smoothing (Li et al., 2018; Oono & Suzuki, 2020); (2) over-squashing and under-reaching (Alon & Yahav, 2020; Topping et al., 2022); and (3) limited expressivity bounded by 1-WL (Xu et al., 2019).

Researchers dedicate relentless efforts to overcome the aforementioned limitations and lead to three research directions: (1). graph Transformers; (2) higher-order GNNs; (3) subgraph GNNs.

### D.3 Graph Positional Encoding and Structural Encoding

Attention mechanisms are structure-invariant operators which sense no structural information inherently. Therefore, Transformers strongly rely on positional encoding to capture the structural information (Vaswani et al., 2017; Dosovitskiy et al., 2021; Su et al., 2024). Designing graph positional encoding is challenging compared to the counterpart in Euclidean spaces, due to the irregular structure and the symmetry to permutation. Widely used graph positional encoding includes absolute PE: LapPE (Dwivedi & Bresson, 2021; Huang et al., 2024); and relative PE: shortest-path distance (Ying et al., 2021), resistance-distance (Zhang et al., 2023b), and RRWP (Ma et al., 2023). Recent works (Black et al., 2024; Zhang et al., 2024b) study the connections between absolute PE and relative PE.

Besides the aforementioned PE, there exist several structural encoding (SE) approaches that aim to enhance MPNNs, e.g., RWSE (Dwivedi et al., 2022b), substructure counting (Bouritsas et al., 2022), and homomorphism counting (Jin et al., 2024). These SEs can effectively improve the empirical performance and/or theoretical expressivity of MPNNs. Although they are not designed specifically for graph Transformers, integrating them into graph Transformers is usually beneficial.

### D.4 Higher-order GNNs

Besides graph Transformers, inspired by the $K$-WL (Weisfeiler & Leman, 1968) and $K$-Folklore WL (Cai et al., 1992) frameworks, $K$-GNNs (Morris et al., 2019; Zhou et al., 2023; Maron et al., 2019) uplift GNNs to higher-order, treating a tuple of $K$ nodes as a token and adapting the color refinement algorithms accordingly. Some other higher-order GNNs (Bodnar et al., 2022; 2021) are less closely related to $K$-WL but still perform well. $K$-GNNs can reach theoretical expressivity bounded by $K$-WL, but are typically computationally costly, with $O(N^K)$ computational complexity.

### D.5 Subgraph GNNs

Subgraph GNNs (Bevilacqua et al., 2022; Frasca et al., 2022; Zhang et al., 2024a; Zhang & Li, 2021; Huang et al., 2023) focus on improving the expressivity of GNNs beyond 1-WL. Unlike graph Transformers and higher-order GNNs, subgraph GNNs typically do not change the model architecture. Instead, MPNNs are trained using a novel learning pipeline: a graph is split into multiple subgraphs, subgraph representations are learned independently, and then the subgraph representations are merged to obtain the graph representation. Node-level subgraph GNNs are typically bounded by 3-WL (Frasca et al., 2022), while the expressivity of edge-level subgraph GNNs is still under-explored. Huang et al. (2023) reveals that the expressivity of edge-level GNNs might partially go beyond 3-WL. Note that subgraph GNNs usually have high memory requirements due to the need to save multiple copies for each input graph.

Although most subgraph GNNs use MPNNs as the base model, the potential integration with stronger graph models, e.g., higher-order GNNs and graph Transformers, is still an open question.

### D.6 PPGT v.s. TokenGT

The early work TokenGT (Kim et al., 2022) establishes a framework for learning graphs with pure Transformers. Despite the significant contribution of Kim et al. (2022), the notion of "pure Transformers" in TokenGT differs substantially from the standard usage in other domains, as it relies on a specialized tokenization scheme that departs from the conventional understanding of "plain" Transformers.

We provide a detailed comparison between PPGT and TokenGT (Kim et al., 2022), focusing on their design principles, expressivity, and computational properties.

**Tokenization and Representation.** TokenGT treats both nodes and edges as tokens, converting a graph with $N$ nodes and $E = DN$ edges into a sequence of $M = N + E$ tokens, where $D$ denotes the average

node degree. This design enables the model to directly incorporate edge and structural information into the Transformer architecture, but also incurs a computational cost that is at least a factor of $D$ higher.

In contrast, PPGT operates purely on node tokens while incorporating edge and structural information through graph positional encodings (PE), e.g., Shortest-path distance, RRWP. This avoids explicitly introducing edge tokens while still capturing rich relational information.

**Attention Mechanism.** TokenGT applies standard full self-attention over all tokens, resulting in interactions between all node–node, node–edge, and edge–edge pairs. While expressive, this leads to a computational complexity of $\mathcal{O}((N + ND)^2) = O(N^2 D^2)$, at least $D^2$ higher computational cost.

PPGT, on the other hand, only performs attention on node tokens like regular Transformers, with structural information injected via graph positional encodings. The attention complexity is remains $\mathcal{O}(N^2)$ as plain Transformers.

**Graph Positional Encoding.** TokenGT relies on a specialized design of graph PE to explicitly account for edge tokens within the Transformer. This design constrains the range of applicable graph PEs in TokenGT. For instance, extending relative graph PEs to TokenGT is non-trivial, as it must consistently handle interactions involving edge tokens.

PPGT, as a plain Transformer, is compatible with a wide range of graph positional encodings (PE), including Laplacian PE, RRWP, and resistance distance, enabling the direct injection of global positional and structural information into attention. In particular, PPGT naturally supports both absolute and relative graph PEs without requiring modifications to the tokenization scheme. This flexibility allows the model to leverage expressive distance-based encodings to capture long-range dependencies efficiently. Empirically, we observe that relative graph PEs, such as RRWP, are especially effective, as they provide fine-grained pairwise structural information that aligns well with the attention mechanism.

**Expressivity.** Theoretically, Kim et al. (2022) show that TokenGT is equivalent to 2-IGN (Maron et al., 2018), and is therefore as expressive as 2-WL. In other words, TokenGT is strictly more expressive than 1-WL, yet remains strictly less expressive than 3-WL.

PPGT falls within the GD-WL framework and, when equipped with suitable graph PE, achieves expressivity strictly stronger than 1-WL while being upper-bounded by 3-WL. Consequently, PPGT attains the same theoretical expressivity range as TokenGT, but with a more favorable computational profile.

Empirically, as shown in Tab. 4, on the large-scale PCQM4Mv2 benchmark, PPGT outperforms TokenGT while using only 36% of its parameters, highlighting its superior empirical efficiency and capacity.

# E   Theoretical Analysis

## E.1   Theoretical Expressivity of PPGT

Our PPGT with s$L_2$ attention (with URPE) and RRWP as graph PE falls within the WL-class of Generalized-Distance (GD)-WL (Zhang et al., 2023b). The expressive power of GD-WL depends on the choice of generalized distance (i.e., graph positional encoding). With a suitable graph PE, such as resistance distance (RD) and RRWP, GD-WL is strictly more expressive than 1-WL and is bounded by 3-WL (1-WL $\sqsupset$ GD-WL $\sqsupset$ 3-WL).

The detailed discussion of GD-WL GNNs and its theoretical expressivity can be found in *Sec. 4* and *Appx. E.3* of Zhang et al. (2023b), as well as in *Sec. 5* of Zhang et al. (2024b). The discussion on RRWP can be found in *Sec. 3.1.1* of Ma et al. (2023). The empirical study of realized expressivity (in Sec. 5.1) also matches the theoretical expressivity of PPGT.

Based on the GD-WL analysis framework (Zhang et al., 2023b), the proof is straightforward. However, we still provide a simple proof for the completeness of the conclusion.

**Proposition E.1.** *Powerful Plain Graph Transformers (PPGT) with generalized distance (GD) as graph PE are as powerful as GD-WL, when choosing proper functions $\phi$ and $\theta$ and using a sufficiently large number of heads and layers.*

For a graph $\mathcal{G} = (\mathcal{V}, \mathcal{E})$, the iterative node color update in GD-WL test is defined as:

$$\chi_{\mathcal{G}}^{\ell}(v) = hash(\{\!\!\{(d_{\mathcal{G}}(v,u), \chi_{\mathcal{G}}^{\ell-1}(u)) : u \in \mathcal{V}\}\!\!\}). \tag{13}$$

where $d_{\mathcal{G}}(v,u)$ denotes a distance between nodes $v$ and $u$, and $\chi_G^0(v)$ is the initial color of $v$. The multiset of final node colors $\{\!\!\{\chi_G^L(v) : v \in \mathcal{V}\}\!\!\}$ at iteration $L$ is hashed to obtain a graph color.

**Lemma E.2.** *(Lemma 5 of Xu et al. (2019)) For any countable set $\mathcal{X}$, there exists a function $f : \mathcal{X} \to \mathbb{R}^n$ such that $h(\hat{\mathcal{X}}) := \sum_{x \in \hat{\mathcal{X}}} f(x)$ is unique for each multiset $\hat{\mathcal{X}} \in \mathcal{X}$ of bounded size. Moreover, for some function $\phi$, any multiset function $g$ can be decomposed as $g(\hat{\mathcal{X}}) = \phi(\sum_{x \in \hat{\mathcal{X}}} f(x))$.*

*Proof of Proposition E.1.* In this proof, we consider shortest-path distance (SPD) as an example of generalized distance (GD), denoted as $d_G^{\mathrm{SPD}}$, which can be directly extended to other GDs such as the resistance distance (RD) (Zhang et al., 2023b) and RRWP (Ma et al., 2023). Note that the choice of GD determines the practical expressiveness of GD-WL.

We consider all graphs with at most $n$ nodes to distinguish in the isomorphism tests. The total number of possible values of $d_G$ is finite and depends on $n$ (upper bounded by $n^2$). We define

$$\mathcal{D}_n = \{d_G^{\mathrm{SPD}}(u,v) : G = (\mathcal{V}, \mathcal{E}), |\mathcal{V}| \leqslant n, u, v \in \mathcal{V}\}, \tag{14}$$

to denote all possible values of $d_G^{\mathrm{SPD}}(u,v)$ for any graphs with at most $n$ nodes. We note that since $\mathcal{D}_n$ is a finite set, its elements can be listed as $\mathcal{D}_n = \{d_{G,1}, \cdots, d_{G,|\mathcal{D}_n|}\}$.

Then the GD-WL aggregation at the $\ell$-th iteration in Eq. (13) can be equivalently rewritten as (See Theorem E.3 in Zhang et al. (2023b)):

$$\chi_G^{\ell}(v) := \mathrm{hash}\left(\chi_G^{\ell,1}(v), \chi_G^{\ell,2}(v), \cdots, \chi_G^{\ell,|\mathcal{D}_n|}(v)\right),$$
$$\text{where } \chi_G^{\ell,k}(v) := \{\!\!\{\chi_G^{\ell-1}(u) : u \in \mathcal{V}, d_G(u,v) = d_{G,k}\}\!\!\}. \tag{15}$$

In other words, for each node $v$, we can perform a color update by hashing a tuple of color multisets determined by the $d_G$. We construct the $k$-th multiset by injectively aggregating the colors of all nodes $u \in \mathcal{V}$ at a specific distance $d_{G,k}$ from node $v$.

Assuming the color of each node $\chi_G^l(v)$ is represented as a vector $\mathbf{x}_v^{(l)} \in \mathbb{R}^C$, by setting the query and key projection matrices $\mathbf{W}_Q, \mathbf{W}_K$ as zero matrices and $\theta$ as a zero function following Zhang et al. (2023b), the attention layer of PPGT with URPE of the $h$-th head (Eq. (5)) can be written as

$$\hat{\mathbf{x}}_v^{(l),h} := \frac{1}{|\mathcal{V}|} \sum_{u \in \mathcal{V}} (\mathbf{W}_O^h \mathbf{W}_V^h \mathbf{x}_u^{(l)}) \cdot \phi^h\big(d_G(u,v)\big). \tag{16}$$

By defining $\phi(d) := \mathbb{I}(d = d_{G,h})$, where $\mathbb{I} : \mathbb{R} \to \mathbb{R}$ is the indicator function, $d_{G,h} \in \mathcal{D}_n$ is a pre-determined condition, we can have

$$\hat{\mathbf{x}}_v^{(l),h} = \frac{1}{|\mathcal{V}|} \sum_{d_G(u,v) = d_{G,h}} \mathbf{x}_u^{(l)}. \tag{17}$$

where $\mathbf{W}_O^h$ and $\mathbf{W}_V^h$ are dropped since they can be absorbed into the following feed-forward networks (FFNs). Note that the constant $\frac{1}{|\mathcal{V}|}$ can be extracted with an additional head and injected back to node representation in the following FFNs.

Then, we can invoke Lemma E.2 to establish that each attention head of PPGT can implement an injective aggregating function for $\{\!\!\{\chi_G^{l-1}(u) : u \in \mathcal{V}, d_G(u,v) = d_{G,h}\}\!\!\}$. The summation/concatenation of the output

from attention heads is an injective mapping of the tuple of multisets $\left(\chi_G^{l,1}, \cdots, \chi_G^{l,|\mathcal{D}_n|}\right)$. When any of the linear mappings has irrational weights, the projection will also be injective. Therefore, with a sufficiently large number of attention heads, the multiset representations $\chi_G^{l,k}, k \in [|\mathcal{D}_n|]$ can be inejectively obtained.

Therefore, with a sufficient number of attention heads and a sufficient number of layers, PPGT is as powerful as GD-WL in distinguishing non-isomorphic graphs, which concludes the proof. □

It is worth mentioning that the expressivity upper bound of GD-WL (i.e., 2-FWL/3-WL), given by Theorem 4.5 in Zhang et al. (2023b), is based on the usage of generalized distance (e.g., resistance distance) as graph PE. If we add PE that provides finer structural information, PPGT might surpass the 3-WL expressivity upper bound. This is empirically verified in Section 5.1, where we demonstrate using I$^2$GNN to generate PE for PPGT.

### E.2 LN and RMSN are Magnitude Invariant

**Proposition E.3.** *LN and RMSN are magnitude-invariant, i.e., for an input vector $\mathbf{x} \in \mathbb{R}^D$ and a positive scalar $c \in \mathbb{R}^+$, $LN(c \cdot \mathbf{x}) = LN(\mathbf{x})$ and $RMSN(c \cdot \mathbf{x}) = RMSN(\mathbf{x})$.*

*Proof of Proposition E.3.* From Eq. (1), we immediately have:

$$
\begin{aligned}
\mathrm{RMSN}(c \cdot \mathbf{x}) &:= \frac{c \cdot \mathbf{x}}{\frac{1}{\sqrt{D}}\|c \cdot \mathbf{x}\|} \cdot \boldsymbol{\gamma}, \\
&= \frac{\cancel{c} \cdot \mathbf{x}}{\frac{\cancel{c}}{\sqrt{D}} \cdot \|\mathbf{x}\|} \cdot \boldsymbol{\gamma}, \text{ since vector norm is absolutely homogeneous and } c > 0 \\
&= \mathrm{RMSN}(\mathbf{x}), \forall c > 0,
\end{aligned}
\tag{18}
$$

which proves the proposition for RMSN. The proof for LN follows the same steps. □

## F Limitations and Discussion

As plain Transformers, PPGTs still require $O(N^2)$ computational complexity like other plain Transformers.

This limits the ability of PPGTs to handle very large-size inputs, which are known as large-scale graphs in the context of graph learning.

Note that the large-scale graph data is a different concept from large-scale graphs. We provide a more detailed discussion here.

### F.1 Large-scale Graph Datasets v.s. Large-scale-graph Datasets

As foundation models become increasingly popular, researchers have grown interested in the models' capacity to learn from massive volumes of data, a.k.a., large-scale data.

However, in the domain of graph learning, persistent confusion remains when discussing scaling up graph models for large-scale datasets, particularly concerning two distinct concepts: "large-scale datasets" versus "large-scale-graph datasets."

We provide a preliminary clarification regarding them:

- **Large-scale (graph) datasets** (a large number of examples, i.e., graphs): This aligns with the conventional understanding of dataset scale in machine learning (e.g., language and vision), where capacity is the crucial factor for learning from the vast volume of training data. This type of dataset is the usual scenario for training large foundation models.

- **Large-scale-graph dataset** (many nodes in a graph): This corresponds to the challenges in long-context learning in language tasks and gigapixel image processing in vision tasks. The challenges mainly lie in the computational efficiency and memory consumption of the models. These datasets are usually small-scale in terms of examples, typically containing only one graph per dataset.

Even though these two research directions are both highly important in solving real-world problems. The improvements on "scalability" and on "capacity" of (graph) models are generally two distinct research directions that are often mutually exclusive in practice.

## F.2 Efficiency Techniques for Plain Transformers

As plain Transformers, PPGTs can potentially adopt several efficiency techniques developed for plain Transformers in graph-related and other domains, directly or with modifications.
BigBird (Zaheer et al., 2020) and Exphormer (Shirzad et al., 2023) propose building sparse attention mechanisms through sparse computational graphs, using random graphs and expander graphs for language tasks and graph learning tasks, respectively. They can be directly adopted by PPGTs.
Performer (Wu et al., 2021) proposes building low-rank attention via positive orthogonal random features approaches, which can be directly applied to $sL_2$ attention. However, it is not directly compatible with relative PE.
Ainslie et al. (2023) proposes grouped-query attention (GQA) to save the memory bandwidth. It is a more specific design for decoder-only Transformers (for KV-cache), and might not introduce remarkable benefits to encoder-based Transformers, such as ViTs (Dosovitskiy et al., 2021; Touvron et al., 2021a; Liu et al., 2021), Time-series Transformers (Nie et al., 2023; Zhang et al., 2024c) and most GTs (Kreuzer et al., 2021; Ying et al., 2021; Zhang et al., 2023b; Ma et al., 2023) including PPGTs.

We mention only a few efficiency techniques for plain Transformers here. Owing to the plain Transformer architecture, PPGTs can potentially adopt efficiency techniques developed in other domains and leverage their training advances.

## F.3 Impact Statement

This paper presents work whose goal is to advance the field of Geometric/Graph Deep Learning. There are many potential societal consequences of our work, none of which we feel must be specifically highlighted here.

