# OpenReview forum: "Plain Transformers Can be Powerful Graph Learners"
_TMLR — Accepted by TMLR_

### Review · Reviewer_xzB3 · 2026-03-04

**Summary Of Contributions:**

This paper proposes a GNN architecture which moves closer to the standard transformer architecture than previous approaches, while also achieving strong performance across several graph metrics. The key aspects to the proposed approach are the adaptive RMS normalization, which allows a model to learn to preserve magnitude along with angle information, the simplified L2 attention, which can leverage key-query length similarity, and the spectral based sinusoidal universal position embeddings, which provide an absolute label for each node within the graph structure. The authors compare prior transformer-based GNNs and show competitive or improved performance, while utilizing a simpler design.

Strengths:

-	The paper is well written and easy to follow.
-	The discussion tends toward the theoretical, with compelling evidence and general applicability beyond GNNs.
-	The ablations support the claims of architecture improvements and are complimented by the graph metric evaluations.

Weaknesses:

-	Most of the paper focuses on the node path but does not adequately describe how the edges and adjacency are handled.
-	The attention results are limited to low-dimensional heads, which weakens claims for the more general case.
-	The proposed method will have a limited maximum graph size due to the $O(N^2)$ scaling of attention.

**Additional Comments:**

Much of the work (including ablations) was performed with low-dimensional attention heads (d=8). Do the results still hold at more standard dimensions such as 32, 64,128? It may be that capturing a polar representation has a bigger effect on the low-dimensional regime, whereas such gains disappear under more standard attention configurations.

Some of the motivating statements may be overclaimed and potentially contradictory. For example, deviations from the standard transformer architecture prevent easy adoption, and that using a standard transformer architecture allows the use of extensively optimized software stacks. However, it seems that PPGT will also exhibit these same pitfalls. Similarly, adjusting the attention mechanism and adding a floating-point mask will not be compatible with highly optimized libraries such as flash attention. The current formulation also assumes that |q|=|k|, which would not be true in the case of cross-attention.

With the AdaRMS approach, were any special considerations necessary to ensure training stability? For example, clamping $\alpha_i\in[0,1]$. What is the typical distribution of these values?

**Audience:**

Yes

**Audience Explanation:**

The paper proposes an interesting combination of architecture modifications, which are simpler than prior graph-transformer architectures. These modifications are well ablated and help demonstrate the impact of each, which may further have implications for general dense transformer networks. Aside from the architectural improvements, the paper contains detailed discussion on the limitations of SDP attention and modeling of representations (magnitude-angle vs angle) which may be relevant to general dense transformers, smaller transformers, and quantized representations.

**Broader Impact Concerns:**

No concerns.

**Claims And Evidence:**

Yes

**Claims Explanation:**

The authors demonstrate matching or improved performance against several competing models on multiple metrics. Additionally, the ablations clearly demonstrate how the proposed architecture improvements impact performance. These ablations include compelling evidence for how AdaRMS and sL2 interact, while the position embeddings help improve performance over the prior baseline.

**Requested Changes:**

The authors should better elaborate on how the edge and adjacency paths are handled within the network. As the paper reads, it seems as if the model treats all nodes as fully connected, ignoring graph structure completely.

---

> ### Author Response · Authors · 2026-04-03
>
> > RC: The authors should better elaborate on how the edge and adjacency paths are handled within the network. As the paper reads, it seems as if the model treats all nodes as fully connected, ignoring graph structure completely.
>
> We thank the reviewer for raising this important point.
>
> While the Transformer backbone indeed operates on a fully connected computational graph, our model does not ignore the underlying graph structure. Instead, structural/positional information is explicitly injected through the
> graph positional encoding (PE). We use RRWP (Ma et al., 2023), as described in Sec. 3.3 (highlighted as [h3]). **We have updated more detailed description on how to inject RRWP into PPGT in Sec. 3.3 (also highlighted as [h3]) in the revised version.**
>
> Concretely, RRWP encodes multi-step transition probabilities between each pair of nodes on the graph, thereby capturing both local and higher-order structural relationships between nodes. These encodings have been shown effective in previous works (Ma et al., 2023; 2024). As a result, although attention is computed with a fully connected computational graphs, it is effectively perceive the graph structure.
>
>
> > Additional Comment 1 - Much of the work was performed with low-dimensional attention heads (d=8). Do the results still hold at more standard dimensions such as 32, 64,128?
>
> Our results are not limited to low-dimensional attention heads. On PCQM4Mv2 (Table 4), PPGT uses a head dimension of 32 (16 heads, hidden size 512), which is within the standard regime, and still outperforms most baselines with fewer parameters.
>
> For the remaining benchmarks, we tune under a fixed 500K parameter budget which is the standard in these benchmarks for fair comparison. Under this constraint, the optimal configurations tend to favor smaller head dimensions due to the trade-off between depth, width, and number of heads, rather than any limitation of the method.
>
> We further conduct scaling experiments on PCQM4Mv2 using a ViT-Base configuration (hidden size 768, 8 heads). Due to computational constraints, training is still ongoing; at epoch 96 (out of 300), PPGT already achieves 0.08608 (17.6M variant was trained 500 epochs.). This result provides additional evidence that PPGT remains effective with higher-dimensional attention heads.
>
> > Additional Comments 2 -- adjusting the attention mechanism and adding a floating-point mask will not be compatible with highly optimized libraries such as Flash Attention.
>
> This is an important observation.
> It is true that FlashAttention 2.0 does not directly support floating-point attention masks (for RRWP as well as the $|K|$ injection). However, unlike MLP-based attention variants, introducing floating-point masks does not fundamentally alter the core
> attention computation mechanism and is therefore compatible with existing highly optimized attention kernels.
>
> In fact, support for floating-point masks is becoming trend: FlashAttention 3.0 and FlexAttention already support ALiBi-style floating-point biases, and xFormers provides support for floating-point masks with full backpropagation.
>
> > Additional Comment 3 -- The current formulation also assumes that $|q|=|k|$, which would not be true in the case of cross-attention.
>
> PPGT does not assume $|q| = |k|$. Instead, it explicitly captures discrepancies between $|q|$ and $|k|$ via the Euclidean distance, which inherently reflects differences in vector magnitudes.
>
> When such discrepancies are not desired, AdaRMSN can enforce $|q| = |k| = \sqrt{d}$ through normalization, consistent with standard LayerNorm (LN) and RMSNorm (RMSN).
>
> > Additional Comment 4 -- With the AdaRMS approach, were any special considerations necessary to ensure training stability?
>
> In our experiments, we did not find it necessary to explicitly clamp $\alpha_i$ values to the $[0,1]$ range, and training was stable without imposing additional constraints.
> However, for particularly challenging training scenarios—such as those involving unusually large learning rates —clamping could be considered as a precautionary measure to further enhance stability.
> On the other hand, enforcing such constraints may potentially restrict the expressive power and flexibility of AdaRMSN.
> **In the revised version, we include a stress test for AdaRMSN (Appx. C.4)**. The results show that, with proper initialization, AdaRMSN exhibits training stability comparable to RMSN in the early stages, even under larger learning rates.
>
> Regarding the typical distribution of $\alpha_i$ values, we did not observe a consistent pattern. The values are adaptively learned from the data and can vary depending on the dataset, model configuration, and initialization. As such, their distribution is data- and context-dependent rather than exhibiting a universal behavior.
>
>
> --------
> - Ma, L., Lin, C., Lim, D., Romero-Soriano, A., K. Dokania, P., Coates, M., H.S. Torr, P., and Lim, S.-N. Graph Inductive Biases in Transformers without Message Passing. ICML 2023

---

### Review · Reviewer_uPq4 · 2026-03-13

**Summary Of Contributions:**

This paper proposes Powerful Plain Graph Transformers (PPGT) to prove that standard Transformer architectures can excel at graph learning with minimal modifications. The authors introduce AdaRMSN to prevent the loss of token magnitude information, a simplified $L_2$ ($sL_2$) attention to better measure token closeness, and a sinusoidal PE enhancement to overcome the spectral bias of neural networks. PPGT achieves theoretical expressivity matching the GD-WL framework and demonstrates superior empirical performance on various benchmarks like ZINC and PCQM4Mv2 while maintaining a simpler, more unified architecture than existing graph-specific models. Key strengths include its simplicity and strong generalization, while its main weakness is the $O(N^2)$ complexity inherent to plain Transformers.

**Audience:**

Yes

**Audience Explanation:**

The findings are highly relevant to researchers seeking to unify deep learning backbones across different modalities, as the paper shows that graphs can be handled without domain-specific message-passing architectures. The insights into how standard normalization and attention mechanisms fail to capture graph-specific data properties like multiset cardinalities are valuable for the broader machine learning community. Additionally, the demonstration that "plain" models can match the expressivity of specialized GNNs offers significant practical benefits for hardware and software optimization.

**Broader Impact Concerns:**

None.

**Claims And Evidence:**

Yes

**Claims Explanation:**

The claims are supported by formal proofs demonstrating that PPGT reaches the theoretical limits of GD-WL algorithms. Ablation studies on the ZINC dataset confirm that each component, such as AdaRMSN, $sL_2$ attention, and SPE, quantifiably improves model performance. Furthermore, a visualization case study directly illustrates that AdaRMSN preserves the magnitude information lost by standard normalization layers. Extensive benchmarking across multiple datasets also shows PPGT consistently matching or exceeding the performance of more complex state-of-the-art architectures.

**Requested Changes:**

The paper notes that PPGT inherits the $O(N^2)$ complexity of plain Transformers, which limits direct applicability to very large graphs. While the authors discuss sampling strategies like Node2Subgraph for OGBN-ArXiv, the manuscript would be significantly strengthened by a more detailed discussion of the trade-off between subgraph size, node coverage, and final accuracy.

In Section 5.1, the authors mention a mismatch between theoretical and empirical expressiveness. It is important to clarify whether the gains from SPE are purely due to easier optimization (mitigating spectral bias) or if there are specific graph structures where the base RRWP fails without sinusoidal enhancement.

Provide a comparison of the number of parameters and training FLOPs against other "efficiency-oriented" Transformers like SGFormer or Exphormer to provide a more complete picture of the efficiency-performance trade-off.

Expand on the sensitivity of the $S$ hyperparameter in SPE across different datasets; while it is analyzed for CFI graphs, its impact on real-world regression tasks like ZINC would be informative.

The discussion on the initialization of AdaRMSN parameters $\alpha$ and $\beta$ is brief. Additional details on the stability of these parameters during the early phases of training across different learning rates would be beneficial for reproducibility.

---

> ### Author Response · Authors · 2026-04-03
>
> > RC 1 -- The manuscript would be significantly strengthened by a more detailed discussion of the trade-off between subgraph size, node coverage, and final accuracy.
>
>
> We thank the reviewer for the insightful feedback.
>
> **We have added a more detailed description of Node2Subgraph in Appx. B.4, along with a sensitivity study on the choice of $K$ in Appx. C.5.**
>
> Note that the effective $K$-hop of each induced subgraph may vary due to the rollback mechanism in Node2Subgraph, and depends on the sparsity of the local structure.
> We refer the reviewer to Appx. B.4 for more details of Node2Subgraph.
>
> We would like to emphasize that Node2Subgraph is not the main focus of this work, and is only used to enable experiments on large-scale graphs.
> More advanced graph sampling techniques might further improve the performance of PPGT on OGBN-ArXiv.
>
>
> > RC 2 -- It is important to clarify whether the gains from SPE are purely due to easier optimization (mitigating spectral bias) or if there are specific graph structures where the base RRWP fails without sinusoidal enhancement.
>
>
> Our experimental results indicate that the gains from SPE primarily stem from improved optimization, i.e., mitigating spectral bias.
> It is because SPE is an enhancement layer on top of RRWP and SPE itself does not introduce additional graph structural information. Rather, it serves as an enhancement on top of RRWP.
>
> **We have added clarification in the revised manuscript (Section 5.1, highlighted as [h5]).**
>
>
>
> > RC 3 -- Provide a comparison of the number of parameters and training FLOPs against other "efficiency-oriented" Transformers like SGFormer or Exphormer to provide a more complete picture of the efficiency-performance trade-off.
>
>
>
> Unfortunately, we are unable to report training FLOPs for all “efficiency-oriented” Transformers, as many works do not disclose exact configurations for each dataset.
> For instance, SGFormer only provides ranges of hyperparameters used during tuning, which prevents precise FLOPs estimation.
>
> That said, we acknowledge that “efficiency-oriented” Transformers—typically based on linear or sparse attention—achieve $\mathcal{O}(N)$ complexity, whereas PPGT has $\mathcal{O}(N^2)$ complexity.
> However, the efficiency gains of linear-attention in graph Transformers comes at a cost. Specifically, such models degenerate to MPNNs with virtual nodes (Cai et al., 2023),, leading to weaker theoretical expressivity.
>
> This also explains that prior “efficiency-oriented” Transformers primarily target node-level tasks on large-scale graphs, driven by efficiency considerations, and rarely report results on graph-level tasks, which are generally more indicative of model expressivity and capacity.
>
> In contrast, our work centers on the expressivity of Plain Transformers in graph learning.
> Accordingly, experiments on node-level tasks over large-scale graphs are presented as supplementary evidence, primarily to demonstrate applicability in such settings rather than as a primary objective, to address the potential concerns from readers.
>
> More broadly, the “efficiency–performance” trade-off is inherently difficult to quantify, as performance depends not only on model capacity (expressivity), but also on generalization as well as the complexity of the tasks.
> For example, on simpler datasets, higher-capacity models may overfit and yield worse performance.
> In our work, we emphasize that plain Transformers can be expressive in graph learning. Accordingly, we primarily evaluate PPGT on graph-level tasks, which demand model expressivity and capacity rather than efficiency.

---

> > ### Author Response · Authors · 2026-04-03
> >
> > > RC 4 -- Expand on the sensitivity of the hyperparameter in SPE across different datasets; while it is analyzed for CFI graphs, its impact on real-world regression tasks like ZINC would be informative.
> >
> >
> > **In the revised version, we add a sensitivity study of SPE for ZINC in Appx.C.1, highlighted as [h8].**
> >
> > It is worth noting that a larger $S$ in SPE is a double-edged sword. Increasing $S$ can better help mitigate spectral bias by amplifying high-frequency signals; however, it can also make the model more susceptible to noise in the training data and lead to overfitting. Therefore, $S$ need to be tuned for each dataset, with regularization adjusted accordingly.
> >
> >
> >
> > > RC 5 --  The discussion on the initialization of AdaRMSN parameters $\alpha$ and $\beta$ is brief. Additional details on the stability of these parameters during the early phases of training across different learning rates would be beneficial for reproducibility.
> >
> >
> > **In the revised version, we add a more detailed discussion on the initialization of AdaRMSN as well as a stress test of AdaRMSN (compared to bad initialization) in Appx. C.4.**
> >
> >
> > Conducted on ZINC dataset within first 50 epochs.
> > ✓ denotes regular training, ✗ denotes training crash (NaN or loss explosion).
> > Bad-AdaRMSN refers to the variant initialized with α = **1** and β = **0**.
> >
> > | Learning Rate | RMSN | AdaRMSN | Bad-AdaRMSN |
> > |--------------|------|---------|-------------|
> > | 2e-3         | ✓    | ✓       | ✓           |
> > | 7e-3         | ✓    | ✓       | ✗           |
> > | 1e-2         | ✗    | ✗       | ✗           |
> > | 7e-2         | ✗    | ✗       | ✗           |
> >
> >
> > The results show that AdaRMSN domonstrates near-parity with RMSN on the stability, whereas the poorly initialized variant exhibits higher sensitivity to learning rates. This confirms that the proposed initialization is meaningful for maintaining stable training dynamics.
> >
> > -----
> >
> > - Cai, C., Hy, T. S., Yu, R., and Wang, Y. (2023). On the Connection Between MPNN and Graph Transformer.
> > In Proc. Int. Conf. Mach. Learn.

---

### Review · Reviewer_5TNL · 2026-03-20

**Summary Of Contributions:**

The paper revisits applying the Transferformer architecture for Graph without integrating message-passing or incorporating sophisticated attention mechanisms. To do that, the paper proposes three improvements: (1) simplified L2 attention; (2) adaptive root-mean-square normalization; (3) MLP-based stem for graph positional encoding. The paper also demonstrates its effectiveness via empirical performance across various graph datasets.

**Audience:**

Yes

**Audience Explanation:**

The paper proposes the Transferformer architecture for Graph without integrating message-passing or incorporating sophisticated attention mechanisms that TMLR's audience will be interested in. Moreover, the paper writing is clear and easy to follow and the experiments are comprehensive.

**Claims And Evidence:**

No

**Claims Explanation:**

- The claim “Plain Transformers Can be Powerful Graph Learners” resembles what is stated in TokenGT (Kim et al. NeurIPS 2022), i.e.,  “Pure Transformers are Powerful Graph Learners”
- Lacking of novelty since the direction explored by TokenGT (Kim et al. NeurIPS 2022)
- Missing to directly compare with the core baseline (TokenGT) since two works are the same approach.
- The comparison with TokenGT should be conducted on the same scale and with the same number of parameters.
- Lacking theoretical complexity or empirical complexity of the method compared to the baselines.

**Requested Changes:**

- Please address the above weakness
- Can the method scale up with a larger number of parameters?
- Section 2.2 discussion of “Limitation of Transformers” can be misunderstood since some limitations are not exactly current limitations of Transformer or have been solved by other techniques (eg: pre-RMSNorm or residual connections). The paper should discuss “Limitation of Transformers when applying to Graph”.
- The paper should discuss Tokenizer compared with TokenGT (Kim et al. NeurIPS 2022) to illustrate the improvement. And other architecture choices such as PE, Norm, MSA should be compared as well to see the actual improvement.
- Section 3.5 is too limited, if there are no new theoretical results or insights, it should be moved to Appendix or refer to other papers.

---

> ### Author Response · Authors · 2026-04-03
>
> > RC1 - W1 -- The claim “Plain Transformers Can be Powerful Graph Learners” resembles what is stated in TokenGT (Kim et al. NeurIPS 2022), i.e., “Pure Transformers are Powerful Graph Learners”; Lacking of novelty since the direction explored by TokenGT (Kim et al. NeurIPS 2022)
>
>
> We respectfully disagree with the reviewer’s statement.
>
> We believe this concern may stem from insufficient discussion of PPGT in comparison to TokenGT. To address this, we have added a **more detailed analysis of PPGT vs. TokenGT in Appx. D.6** of the revised manuscript.
>
> In brief, TokenGT adopts an unconventional tokenization scheme, where both nodes and edges are treated as tokens. This design is tightly coupled with its specialized positional encoding. In contrast, conventional Transformers treat only nodes as tokens, with edges serving as structural information typically encoded via positional encoding.
> While treating edges as tokens may offer certain benefits, it also introduces architectural constraints that hinder the adoption of many established Transformer components, such as relative positional encoding.
> Therefore, equating TokenGT with PPGT is not technically accurate, and the claim of lacking novelty does not hold.
>
> **More importantly, as demonstrated in our direct comparison (Table 4) on the large-scale PCQM4Mv2 benchmark, PPGT outperforms TokenGT while using only $36\%$ of its parameters, highlighting its superior empirical efficiency and capacity.
> This provides strong evidence that the design of PPGT is indeed more powerful.**
>
>
>
> > RC1 - W2 -- Missing to directly compare with the core baseline (TokenGT) since two works are the same approach.; The comparison with TokenGT should be conducted on the same scale and with the same number of parameters.
>
>
> In our original manuscript, we indeed provide a direct comparison with TokenGT on the **PCQM4Mv2** benchmark (see Table 4 on Page 11).
> Notably, PPGT outperforms TokenGT at a margin while using only $36\%$ of its parameters, demonstrating substantially improved parameter efficiency and architectural capacity.
>
> Furthermore, PPGT exhibits more favorable computational complexity theoretically.
> TokenGT has time complexity $O(N^2 D^2)$, whereas PPGT scales as $O(N^2)$, where $D > 1$ denotes the average node degree. This results in significantly better scalability in practical graph settings.
>
> Together, these results constitute a fair and meaningful comparison, indicating that PPGT achieves superior performance (i.e., more powerful) under a more efficient parameter and computational budget.
>
>
>
> > RC3 - W3 -- Lacking theoretical complexity or empirical complexity of the method compared to the baselines.
>
>
> To address the concerns of reviewer, **we have added the theoretical complexity of PPGT and other major baselines in Appendix C.7 (highlighted as [h4]).**
>
> In our original manuscipt, we also provided an empirical runtime and GPU memory analysis in Appx. B.4 (Appx. C.6 in the revised version).
>
>
> > RC 2 -- Can the method scale up with a larger number of parameters?
>
> Driven by the same computational complexity,
> our method demonstrates scalability comparable to other expressivity-oriented baselines such as GraphGPS, Graphormer, and GRT.
>
> As shown in Table 4 (Page 11), we scale up PPGT to 17.6M parameters, achieving competitive performance compared to even larger models.
>
> To better address the concern of the reviewer.
> We have conducted experiments training an 87M-parameter PPGT (ViT-Base scale)  and a 309M-parameter PPGT (ViT-Large scale) on PCQM4Mv2 from scratch.
> However, due to the model scales and limited computational resources from our end, training has not yet completed.
> Preliminary results show that the 87M-models achieves a test MSE of $0.0860$ at epoch 94 and the 309M-model reaches $0.108$ at epoch 46, under a training schedule of 500 epochs (consistent with the setting of 17.6M version).
> We would update the final results once the experiments are finished.
>
> Note that scaling model parameters is constrained not only by the model architecture itself, but also by the scale of the training dataset.

---

> > ### Author Response · Authors · 2026-04-27
> >
> > For the 87M-parameter PPGT (ViT-Base scale), the model achieves a final validation MAE of 0.08396 (the valid set is used as the test set in the pipeline following GraphGPS).
> >
> > We note that the current optimizer hyperparameters remain suboptimal, as evidenced by multiple loss spikes during training. Further hyperparameter tuning is likely to yield additional performance gains.

---

> ### Author Response · Authors · 2026-04-03
>
> > RC 3 -- Section 2.2 discussion of “Limitation of Transformers” can be misunderstood since some limitations are not exactly current limitations of Transformer or have been solved by other techniques (eg: pre-RMSNorm or residual connections). The paper should discuss “Limitation of Transformers when applying to Graph”
>
>
>
> We thank the reviewer for this insightful comment. We will revise the section title accordingly to clarify that the discussion specifically concerns the limitations of Transformers when applied to graphs.
>
> However, we would like to emphasize that the issues highlighted in Section 2.2 are not fully resolved in Language domains.
> For example, pre-RMSNorm can still discard magnitude information.
> In the revised version, we add a summarization to indicate it in the end of section 2.2 (highlighted as [h7])
>
> However, we acknowledge that these limitations may be less pronounced in other domains, such as language and vision.
>
>
>
> > RC 4 -- The paper should discuss Tokenizer compared with TokenGT (Kim et al. NeurIPS 2022) to illustrate the improvement. And other architecture choices such as PE, Norm, MSA should be compared as well to see the actual improvement.
>
>
> We thank the reviewer for pointing this out.
>
> **As noted in our response to W1, we have added a more in-depth discussion of PPGT vs. TokenGT in Appendix D.6 of the revised manuscript.**
>
> In brief, TokenGT treats both nodes and edges as tokens, whereas PPGT follows the standard Transformer formulation and treats only nodes as tokens.
>
> TokenGT requires specially designed positional encodings to distinguish between node and edge tokens. In contrast, PPGT can directly leverage widely used graph positional encodings; in particular, we adopt RRWP from GRIT (Ma et al., 2023).
> Notably, RRWP cannot be directly incorporated into TokenGT due to fundamental architectural differences, primarily stemming from its specialized tokenization scheme.
>
> Beyond tokenization, TokenGT employs LayerNorm and standard dot-product attention. In contrast, PPGT, motivated by our analysis, adopts AdaRMSN and $L_2$ attention—a variant of dot-product attention that remains compatible with standard implementations.
>
> In terms of the expressive power, in our direct comparison on the large-scale PCQM4Mv2 benchmark, PPGT outperforms TokenGT while using only $36\%$ of its parameters, highlighting its superior empirical efficiency and capacity.
> This provides strong evidence that the design of PPGT is indeed more powerful.
>
>
> > RC 5 -- Section 3.5 is too limited, if there are no new theoretical results or insights, it should be moved to Appendix or refer to other papers.
>
>
> The theoretical analysis of PPGT is not provided in prior work. Therefore, we believe it remains necessary to include a rigorous proof in the manuscript.
>
> More importantly, **in the original manuscript, the majority of the theoretical discussion was already presented in Appendix E.1.**
> Section 3.5 primarily serves as a high-level pointer to this detailed analysis, and thus should be retained in the main text.

---

### Decision · Action_Editor_BWfg · 2026-05-16

**Recommendation:** Accept as is

**Audience:**

Yes

**Audience Explanation:**

The question addressed in this paper is clearly of interest to the TMLR audience: how far one can go in graph learning while retaining a Transformer architecture close to the standard backbone. The work appears relevant to researchers in graph representation learning and Transformer architectures.

The paper provides also useful insights into other aspects that can be of interest to readers who do not directly work on graph Transformers, such as the trade-offs between architectural simplicity, expressivity, and scalability.

**Claims And Evidence:**

Yes

**Claims Explanation:**

The central claim of this paper is that a Transformer architecture that remains close to the standard dense-attention backbone, with a small number of principled modifications, can be an effective graph learner. This claim is supported theoretically to some extent and empirically.

Two reviewers requested further clarifications about the scope of the efficiency claims in terms of scalability and computational complexity, which were addressed satisfactorily through additional comparisons and expanded discussion. In addition, one reviewer questioned the novelty and adequacy of the comparison to TokenGT, which was also addressed satisfactorily. Other minor points were also adequately addressed.